# Clinical relevance of molecular characteristics in Burkitt lymphoma differs according to age

Birgit Burkhardt [1,24✉], Ulf Michgehl [1,24], Jonas Rohde [1,24], Tabea Erdmann [2], Philipp Berning [2], Katrin Reutter[1], Marius Rohde [3], Arndt Borkhardt [4], Thomas Burmeister [5], Sandeep Dave [6], Alexandar Tzankov [7], Martin Dugas[8], Sarah Sandmann [9], Falko Fend [10], Jasmin Finger[1], Stephanie Mueller[1], Nicola Gökbuget[11], Torsten Haferlach [12], Wolfgang Kern[12], Wolfgang Hartmann [13], Wolfram Klapper [14], Ilske Oschlies [14], Julia Richter[14], Udo Kontny [15], Mathias Lutz [16], Britta Maecker-Kolhoff [17], German Ott[18], Andreas Rosenwald[19], Reiner Siebert[20], Arend von Stackelberg [21], Brigitte Strahm [22], Wilhelm Woessmann [23], Martin Zimmermann[17], Myroslav Zapukhlyak[2], Michael Grau [2,25] & Georg Lenz [2,25]

While survival has improved for Burkitt lymphoma patients, potential differences in outcome between pediatric and adult patients remain unclear. In both age groups, survival remains poor at relapse. Therefore, we conducted a comparative study in a large pediatric cohort, including 191 cases and 97 samples from adults. While *TP53* and *CCND3* mutation frequencies are not age related, samples from pediatric patients showed a higher frequency of mutations in *ID3, DDX3X, ARID1A* and *SMARCA4*, while several genes such as *BCL2* and *YY1AP1* are almost exclusively mutated in adult patients. An unbiased analysis reveals a transition of the mutational profile between 25 and 40 years of age. Survival analysis in the pediatric cohort confirms that *TP53* mutations are significantly associated with higher incidence of relapse (25 ± 4% versus 6 ± 2%, p-value 0.0002). This identifies a promising molecular marker for relapse incidence in pediatric BL which will be used in future clinical trials.

A full list of author affiliations appears at the end of the paper.

Burkitt lymphoma (BL) and its leukemic manifestation Burkitt leukemia (B-AL) is the most common subtype of non-Hodgkin lymphoma (NHL) in children and adolescents accounting for ~50% of cases. In comparison, BL only accounts for 1% of adult NHL cases in Europe and Northern America[1–4]. The genetic hallmark of BL is a translocation involving the *MYC* oncogene and the immunoglobulin loci, primarily immunoglobulin heavy chain (IGH) in 80% of cases [t(8;14)(q24;q32)]. Variant *MYC* translocations involve the light chain loci [t(8;22) or t(2;8)][5]. Aside from *MYC* translocations, somatic single-nucleotide variants, insertions, and deletions (SNV/indels) of the *ID3-TCF3-CCND3* pathway represent the most frequent genetic events in BL, with up to 90% in pediatric cases, but significantly lower frequency in adult BL[6,7]. Recent reports have advanced the molecular understanding of BL biology by identifying additional genetic mutations, revealing genomic and transcriptomic alterations contributing to MYC dysregulation, and tracing the clonal evolution of BL[8–12]. But data on the effect of patient age on the molecular characteristics of BL are very limited and show inconclusive results[7,13].

With current risk-adapted chemoimmunotherapy, the event-free survival exceeds 90 or even 95% for pediatric cohorts[14]. In contrast, 80% of adult BL patients are cured with multi-agent chemotherapy, and survival rates for younger adults are better[15–18]. Disease relapse usually occurs shortly after the end of therapy and is associated with a dismal prognosis and survival rates below 30%[14,19–21].

Given the poor outcome for patients who relapse, there is a critical medical need for treatment optimization. To this end, targeted approaches and risk-based adaptation of treatment intensity is required. To enable such progress in patient care, translational results based on molecular markers and clinical data are urgently needed. Here, we provide a thorough analysis of the mutational profile of 191 pediatric BL/B-AL by sequencing 134 predefined genes in all samples and by investigating somatic copy number aberrations (SCNAs) in a sub-group of 72 samples. Clinical data were sourced for all 191 cases. These results were compared to the mutational landscape of 97 adult BL cases as there is only limited information about the differences of both age groups[22]. Our analyses identified prognostically relevant genetic lesions of *TP53* and *GPC5/MIR17HG* in pediatric instances, and pinpointed promising targets for BL treatment development in future studies.

## Results

**Clinical characteristics and cohort overview**. We evaluated 288 patients with confirmed diagnoses of BL or Burkitt leukemia (B-AL or Burkitt lymphoma with a blast count of 25% or more in the bone marrow). In particular, cases with 11q aberration have been excluded, as they are considered a different entity[23]. For the 191 pediatric cases, the median age at diagnosis was 9 years. 30% of pediatric patients were female. Approximately 10% were diagnosed with stage I/II disease, according to the St. Jude staging system, while about 40% had stage III and 50% stage IV disease. The clinical characteristics of the pediatric cohort are summarized in Supplementary Table 1. The adult cohort comprised 97 cases, with a median age of 52 years at diagnosis, and was 40% female. (Supplementary Data 1). For sample source information, see Supplementary Table 2.

**Mutational landscape of pediatric BL**. We performed targeted next-generation sequencing for fresh-frozen samples, including 86 matched normal tissue for the 191 pediatric cases. To maximize sensitivity and specificity, our analysis pipeline for variant discovery and filtering makes use of several methods and external databases (see Supplementary Fig. 1 and "Methods"). Overall, we identified 1399 SNV/indels for the 191 pediatric cases and 824 SNV/indels in the adult cohort (Supplementary Data 2), representing 7.3 and 8.5 alterations per sample, and ranging from 0 to 23 total SNV/indels per case (Supplementary Data 3). To test the sensitivity of our pipeline, we additionally performed validation experiments. Only two variants detected by Sanger sequencing in validation regions were not discovered by targeted DNA sequencing, in both cases due to deletions that are likely too long for DNA sequencing to recognize (see validation by Sanger sequencing in "Methods" and validation overview in Supplementary Data 4). As another control of our analysis pipeline, we used ten cell lines to verify the concise calling of all SNV/indels (Supplementary Fig 2).

In the full cohort (pediatric and adult cases combined), the highest frequency of SNV/indels was detected in the *MYC* gene, with 479 SNV/indels, resulting in 1.7 SNV/indels per case on average, followed by *ID3* with 1.1 (Supplementary Data 2). Interestingly, *MYC* showed a higher mutation rate in adults when compared to children, at 1.9 and 1.5 per case, respectively (Supplementary Fig. 3). Most of the recurrently mutated genes are involved in regulatory processes and are functionally linked to transcription factor binding (*DDX3X*, *ID3*, *SMARCA4*, *TCF3*, and *TP53*, Supplementary Table 2) or activation of transcriptional processes (*ARID1A*, *SMARCA4*, *TCF3*, *TFAP4*, *FOXO1*, and *TP53*). Furthermore, *DDX3X*, *FOXO1*, *ID3*, *TFAP4*, and *TP53* are also involved in the regulation of apoptotic processes.

Focusing on genes recurrently mutated in at least 10% of cases, 14 were identified in the pediatric cohort (Fig. 1A) and 17 in the adult cohort. To test whether these genes were positively selected by mutation processes, we applied dNdScv[24]. All genes except one showed a dN/dS ratio significantly higher than one, indicating excess coding mutations (Supplementary Data 5). TCF3 was likely not verified only due to alternative transcripts (see "Methods"); mutations were called by strongest consequence logic for TCF3 protein variant 2 (NP_001129611). Interestingly, these recurrently altered genes were significantly different between the two age groups, with *YY1AP1*, *BCL2*, *BTG2*, and *CREBBP* recurrently altered in adult but not pediatric BL cases, while alterations in *GNA13* were more predominant to the pediatric cohort (Fig. 1B). In the pediatric cohort, eight genes (*ID3*, *MYC*, *TP53*, *CCND3*, *SMARCA4*, *DDX3X*, *ARID1A*, and *FBXO11*) were altered in more than 20% of all patient samples, while in the adult cohort only five genes (*MYC*, *TP53*, *CCND3*, *ID3*, and *SMARCA4*) were mutated in 20% of cases or more.

**Recurrent somatic copy number alterations in pediatric BL**. Using single-nucleotide polymorphism arrays, we analyzed 72 pediatric BL cases for SCNAs via GISTIC 2.0 (Fig. 2 and Supplementary Data 6). In all, 4% of all samples were identified as polyploid according to the ASCAT analysis (Supplementary Data 7). Chromosomes 1q (4%), 12p (4%), 12q (4%), and 13q (4%) were mostly affected by arm level amplifications. Remarkably, deletions were detected less frequently in our analysis. For a specific region that only contains the protein-coding gene *GPC5* (chr13:91398619-92867237), we identified recurrent gains (8.4%) or amplification (9.5%) in 18% of patients (Fig. 2). Investigating this locus in detail revealed that the directly preceding miR17-92 cluster is also affected by these copy gains or amplifications (*MIR17HG* at chr13:91347820-91354575, which was shown before; cluster comprised of *MIR17*, *MIR18A*, *MIR19A*, *MIR20A*, *MIR19B1*, and *MIR92A1*; Supplementary Data 7; Fig. 2)[25]. Likewise, four long intergenic non-protein-coding RNAs were affected (*LINC01049*, *LINC00410*, *LINC00380*, and *LINC00379*; chr13:90493288-91211698). On chromosome 1, we identified a

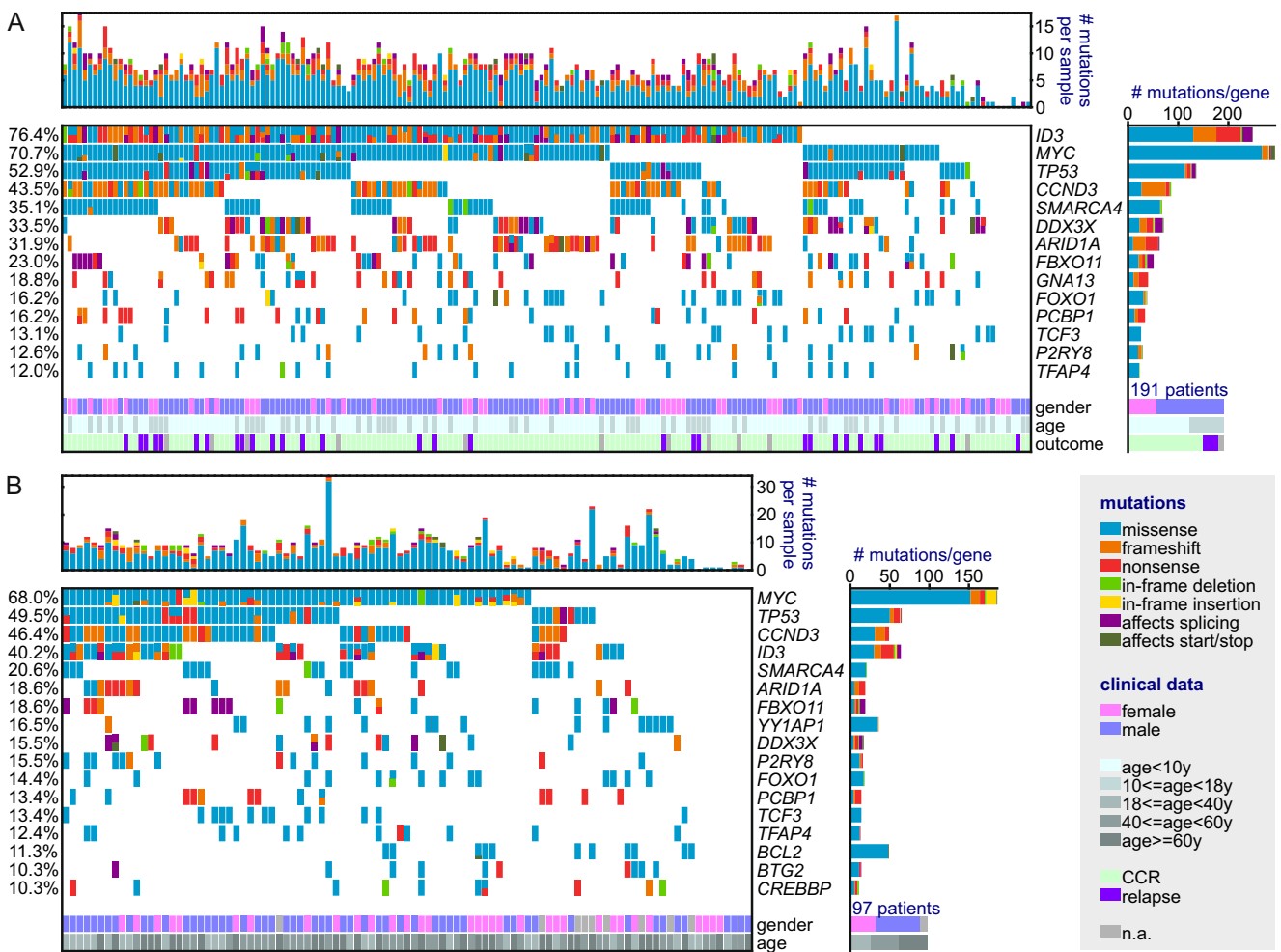

**Fig. 1 Mutational landscape of pediatric and adult BL.** In total, 191 pediatric and 97 adult patient samples were analyzed. Mutations with ≥10% cohort frequency are displayed. **A** In total, 14 recurrently mutated genes were found in the pediatric cohort, with *ID3* being the most frequently altered gene. The color code indicates mutation types or clinical data classes (see legend). **B** In the adult cohort, 17 recurrently mutated genes were detected. In particular, *YY1AP1* and *BCL2* were unmutated or less frequently mutated in the pediatric cohort.

region that was deleted in 8% of all pediatric patient samples, covering *E2F2* and *ID3*. In addition, we detected deletions in 10% of the pediatric cohort of *BLK* (8p23.1) and *CLECL1* (12p13.31) that are exclusively expressed in B cells.

**Recurrent SNV/indels of *ID3* are more frequent in the pediatric cohort.** *ID3* was found to be mutated in 76% of all pediatric cases, but only in 40% of adult samples (*P* = 2e-9, Supplementary Data 8, Figs. 1, 3A), which implies a more important role for *ID3* in the biology of BL/B-AL in children[6,26]. Similarly, SNV/indels affecting the ID3-TCF3-CCND3 pathway were more frequent in children than in adults (87% versus 63%, *P* = 2e-6; Supplementary Fig. 4A, B). Hotspot SNV/indels of *ID3* were similar between pediatric and adult BL/B-AL, mostly located within the HLH domain (aa 42–85) and mainly characterized as missense, nonsense, or frameshift SNV/indels (Supplementary Data 9 and Supplementary Fig. 4C). The most frequent SNV/indel L64F in ID3, was found in 21% of pediatric samples and in 12% of adult cases, followed by P56S, identified in 10% of pediatric patients and 4% of adults. Both are thought to act as loss-of-function mutants and contribute to an increased expression of TCF3, as well as subsequent CCND3 activation (Supplementary Fig. 4D, E)[6,10,26]. *CCND3* itself displayed a frameshift mutation hotspot R271Pfs*76 detected in 15% of pediatric and 6% of adult cases.

**BAF (SWI/SNF) complex components ARID1A/SMARCA4 are recurrently mutated in pediatric Burkitt lymphoma.** The genes encoding ARID1A and SMARCA4 are two components of the BAF complex responsible for gene regulation. They were found to be recurrently mutated in both age groups. However, the mutation rate was significantly higher in the pediatric cohort, with 32% for *ARID1A* and 35% for *SMARCA4*, compared to the adult cohort, with SNV/indels in 19% and 21%, respectively (*P* = 0.008 for *SMARCA4*; *P* = 0.010 for *ARID1A*, Fig. 3A). Overall, *SMARCA4* SNV/indels were mostly missense SNV/indels (84/89, 94%) and located in the DEAD-box (aa 750–942, 33%) or helicase (aa 1110–1194, 31%) region responsible for the alteration of chromatin structure in the process of transcriptional regulation. One *SMARCA4* hotspot was identified with a mutation frequency of 3% in the pediatric cohort (G1232S, Supplementary Data 9). *ARID1A* showed predominantly nonsense (pediatric: 40%, adult: 42%) and frameshift (40%, 31%) SNV/indels, most of which were unique despite the identification of D1850Gfs*4 and S11Afs*91 in four and two pediatric samples, respectively. Only 3% of pediatric cases showed SNV/indels in both genes, while 64% of the pediatric cohort had a mutation in one of the genes. In the adult cohort, 41% of samples were affected by mutually exclusive SNV/indels of either *ARID1A* or *SMARCA4* (*P* = 6e-5 for pediatric versus adult, Supplementary Fig. 5A, B).

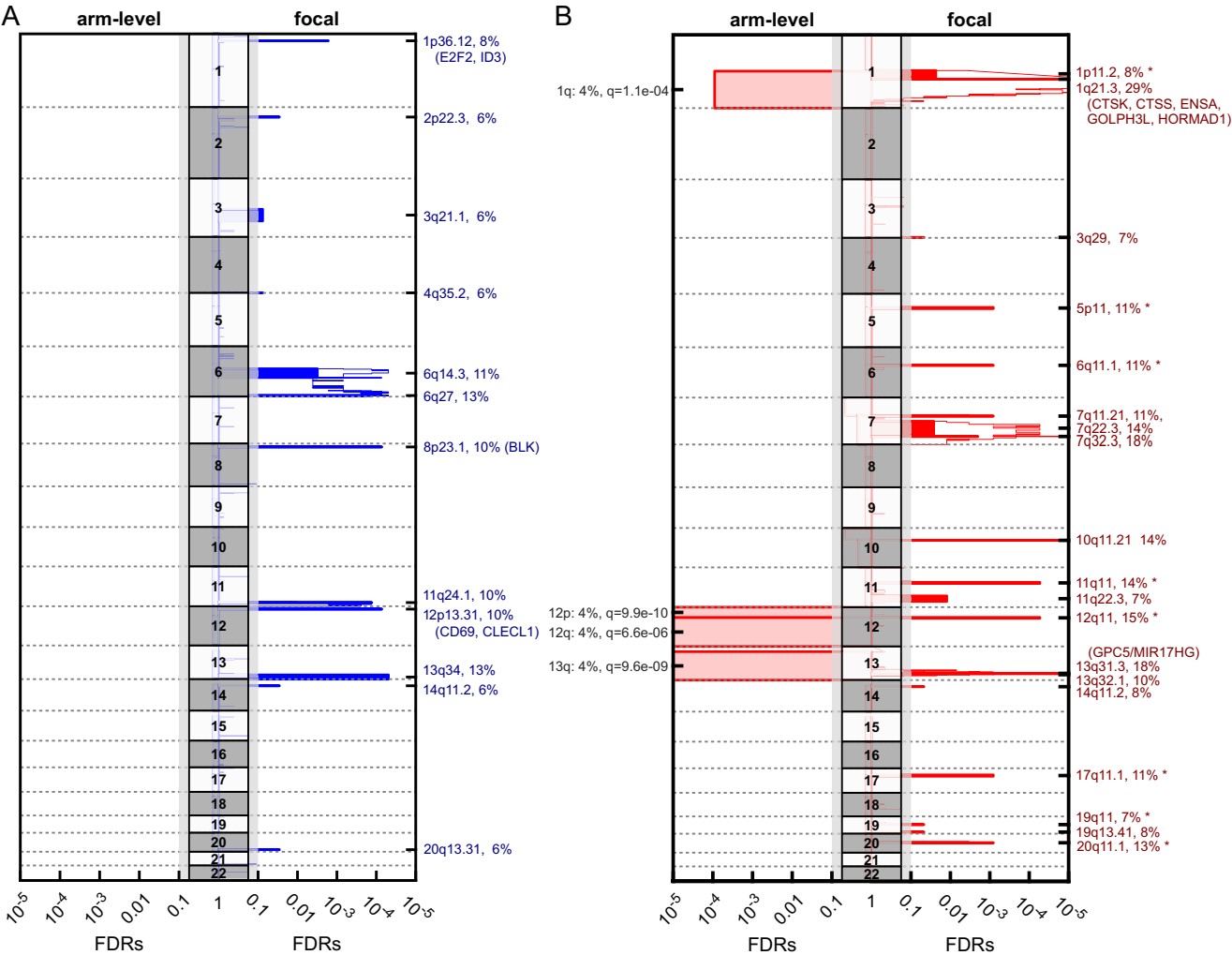

**Fig. 2 Somatic copy number aberrations (SCNAs).** Significant recurring SCNAs are displayed as blocks, based on 72 pediatric cases (55 BL and 17 B-AL cases). Curves indicate residual *q*-values by GISTIC 2.0 after explaining broad aberrations (Supplementary Data 7). **A** Focal deletions affecting *E2F2* and *ID3* were detected. **B** The most consistently observed SCNA contains *GPC5* (*q* = 1.8e-9). Notably, the non-protein-coding *MIR17HG* cluster directly precedes this gene and is also affected by these gains and amplifications (Supplementary Data 6). *: might represent benign (germline) copy number variants.

***DDX3X* SNV/indels are predominantly identified in pediatric Burkitt lymphoma.** Dead box helicase 3 (*DDX3X*) encodes an X-linked RNA helicase found in the nucleus as well as in the cytoplasm and is responsible for transcriptional activation and translation processes[27]. In total, 34% of all pediatric patient samples harbored at least one mutation in this gene compared to 15% in the adult cohort (Fig. 3A, *P* = 7e-4). Most cases of the complete cohort had a single SNV/indel in *DDX3X*, six were double mutated and one case had a triple mutation. In total, 13 SNV/indels resulted in truncated proteins, 16 affected splice sites, and 14 introduced frameshift mutations, ultimately resulting in truncated DDX3X with missing functional DEAD-box or helicase domains. Most of the SNV/indels were localized in the helicase domain, which could potentially lead to a loss-of-function phenotype (Supplementary Fig. 6A).

***GNA13* SNV/indels are more frequent in pediatric Burkitt lymphoma.** We found a higher frequency of *GNA13* alterations in the pediatric cohort compared to the adult cohort (Fig. 3A, 19% versus 7%, *P* = 0.0055). Overall, 48% of all SNV/indels in the pediatric cohort were nonsense SNV/indels with truncation mutations early in the N-terminal region of the protein, including

eight SNV/indels stopping at Q27/28, which suggests loss-of-function mutations (Q27*: 4%/6% mutation frequency, Supplementary Data 9, Supplementary Fig. 6B). Mutual exclusivity of *GNA13* and *P2RY8*, as reported for DLBCL samples, could not be confirmed in our total cohort as we found SNV/indels in both genes[28,29]. While 29% of the pediatric patient samples showed SNV/indels in *GNA13* and/or *P2RY8*, the adult cohort had a slightly lower mutation rate at 21% (*P* = 0.11, cf. Supplementary Fig. 6C, D). The majority of *P2RY8* SNV/indels were located within the transmembrane domain. Loss of function due to misfolded or mislocalized protein appears likely, which has previously been demonstrated for several point mutations (Supplementary Fig. 6E)[28].

***YY1AP1* is predominantly mutated in adult BL.** Recurrent SNV/indels in *YY1AP1* were found in 16% of cases in the adult cohort. The mutation frequency was significantly higher compared to the pediatric cohort (2%, *P* = 1e-5). YY1AP1 is responsible for transcriptional activation, DNA repair and replication, and is linked to other cancer types[30]. We found three hotspot SNV/indels with mutation frequencies ranging from 8-12% directly adjacent (G23R, V24G, S25A) or in close proximity

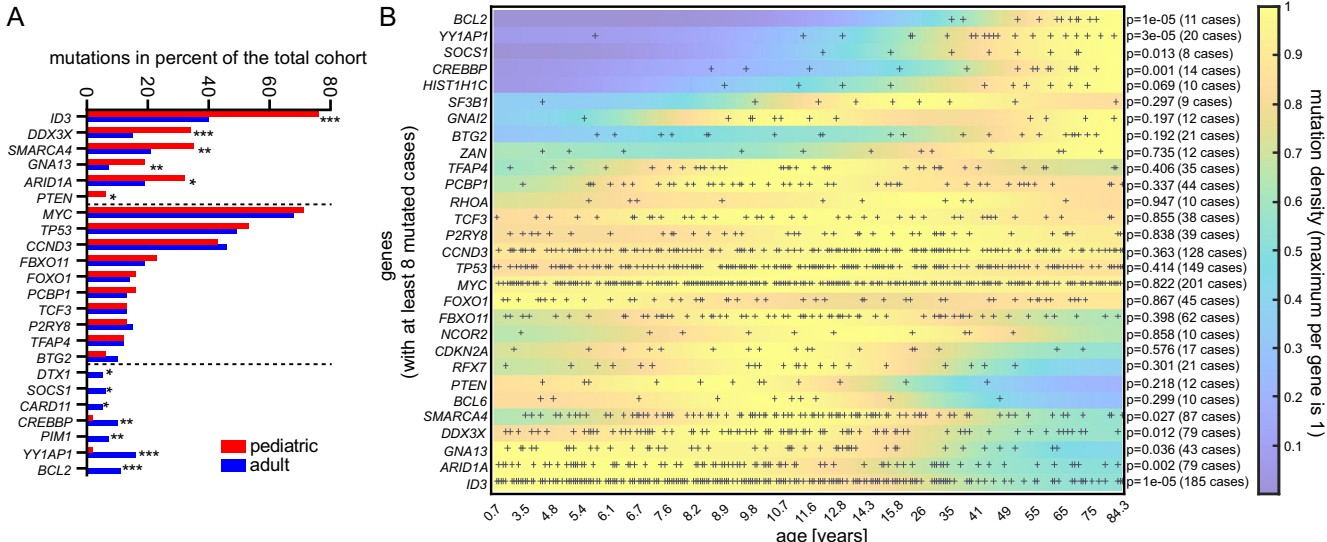

**Fig. 3 Age dependency of the mutation profile. A** Comparison of mutation counts between age groups, using the usual age cutoff of 18 years for adult patients. Mutations in *ID3, DDX3X, SMARCA4, GNA13, ARID1A,* and *PTEN* are significantly more frequent in pediatric patient samples, while *BCL2, YY1AP1, PIM1, CREBBP, CARD11, SOCS1,* and *DTX1* are significantly more frequently altered in the adult cohort (*P* values by one-sided exact Fisher tests, see Supplementary Data 8 for exact *P* values and FDRs, *$P \leq 0.05$, **$P \leq 0.01$, ***$P \leq 0.001$). **B** A cutoff-free enrichment analysis of mutation densities over age largely confirms cutoff-based results but reveals that the main biological transition of the mutational profile occurs later between 25 and 40 years (*P* values by one-sided enrichment analysis and permutation test with 1e5 permutations per tail).

(S30R, 4%), which were exclusively in the adult cohort (Supplementary Data 9). Interestingly, seven of these samples showed a single mutation, while five cases had two or more SNV/indels within these sites. Additional SNV/indels of the S30R site occurred in four cases (Supplementary Data 9).

A second gene with alterations predominantly occurring in the adult cohort was *BCL2*, which was found to be mutated in 11% of the adult BL cohort, while there were no SNV/indels detected in the pediatric cohort (Fig. 3A, *P* = 4e-6). Furthermore, the adult cohort showed significantly higher mutation frequencies in *PIM1* (7%), *CREBBP* (10%), *CARD11* (5%), *SOCS1* (6%), and *DTX1* (5%, Fig. 3A). All of these were either undetected or detected at only very low rates in the pediatric cohort. Taken together, our data reveal a distinct difference between the mutational spectrum in pediatric and adult Burkitt lymphoma with respect to specific genes.

**Cutoff-free mutation density analysis reveals the main transition of the mutational signature occurs between 25 and 40 years.** In addition to the usual age cutoff for adults at 18 years, we performed a cutoff-free mutation enrichment analysis (Fig. 3B). Enrichment tests (see "Methods") confirmed the significance of age biases in genes determined by the Fisher tests above. However, gene mutation densities over age revealed that the biological main transition of the mutational signature surprisingly occurs later, between 25 and 40 years with a median of about 35 years in our study. In particular, *ID3* mutation density relative to children falls to about 40% during this transition, and *BCL2* mutations did not occur before. In addition, a tendency for lower mutation frequencies in some genes can be seen for children younger than 7 years, including *SMARCA4, FBXO11,* and *PCBP1*. In contrast, several other genes including *MYC* and *TP53* did not show any age bias with respect to mutation density. The highest homogeneity of the mutation profile is visible in the age range of about 7–15 years.

**Associations of distinct SNV/indels with clinical characteristics of Burkitt lymphoma.** Associations of genetic markers with

clinical data of pediatric Burkitt lymphoma are still scarce and are urgently needed to identify ways for therapy improvement. To this end, we systematically tested for such associations (Supplementary Data. 8).

Only tendencies to slightly higher rates of SNV/indels in *TP53* (*P* = 0.044) and *SMARCA4* (*P* = 0.01) were observed in adolescents of 10–18 years compared to younger children (Fig. 4A). This is consistent with our mutation over age analysis suggesting a homogeneous mutation signature from about 7–15 years of age.

Strikingly, *DDX3X* showed a very strong bias to male patients, of 46%:4% incidence compared to females (Fig. 4B, *P* = 4e-10). *CCND3* SNV/indels were significantly more frequent in cases with bone marrow (BM) involvement or Burkitt leukemia (Fig. 4C, *P* = 0.003). In contrast, SNV/indels in *GNA13* showed an incidence of only 4%:32% in B-AL, i.e., was significantly less frequent (Fig. 4D, *P* = 6e-7). Likewise, cases with SNV/indels in *GNA13* were less likely to have CNS involvement at diagnosis (Fig. 4E, 5%:23%, *P* = 0.007).

As the information on clinical parameters for the adult patients was limited, the analysis was restricted to differentiation by sex. Again, a male predominance for SNV/indels in *DDX3X* was observed, but only by 3%:22% and with a relatively high false discovery rate due to lower case numbers (Fig. 4F, *P* = 0.006).

***TP53* mutation status is associated with inferior outcome in pediatric patients.** The overall survival for pediatric patients with SNV/indels in *TP53* was significantly inferior to patients with *TP53* wild-type (Figs. 5A, 6A). To exclude effects from other events (e.g., treatment-related deaths, etc.) further analyses focused on the cumulative incidence of relapse. SNV/indels of *TP53* were significantly associated with a cumulative incidence of progression or relapse of 25 ± 4%, compared to 6 ± 2% for BL with *TP53* wild-type status (Fig. 5B, *P* = 0.0002). Remarkably, a higher number of *TP53* SNV/indels per case was not correlated with a higher cumulative incidence of relapse (Supplementary Fig. 7A). The pediatric cohort comprised 40 patients, who received rituximab in addition to chemotherapy according to the NHL-BFM treatment protocol. The prognostic relevance of *TP53*

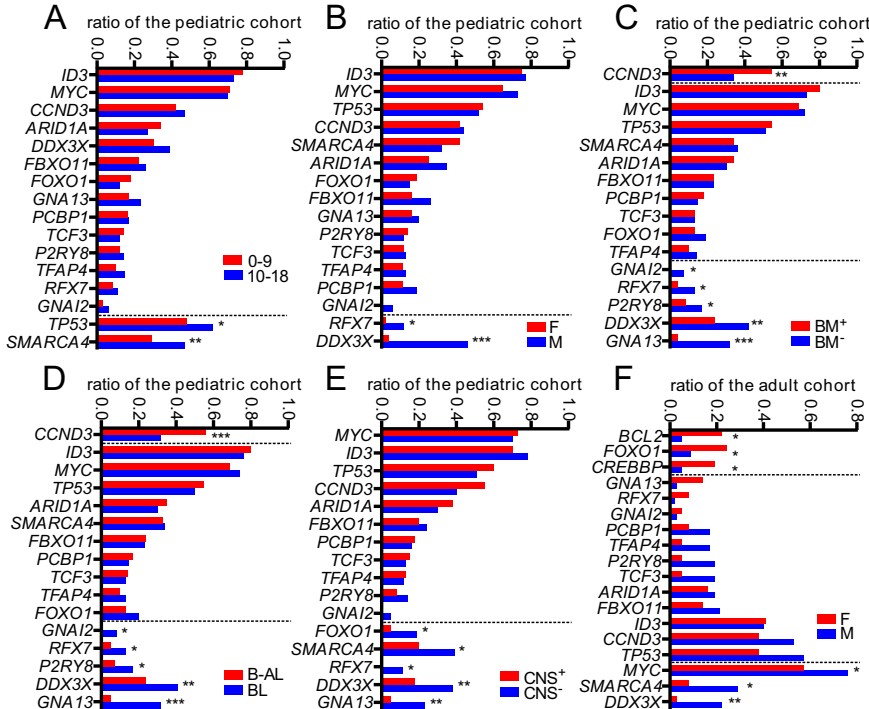

**Fig. 4 Mutational frequencies associated with clinical characteristics.** For the pediatric cohort, gene mutation rates were associated with the following clinical characteristics: age dependency (**A**); sex (**B**); BM (bone marrow, **C**); B-AL vs. BL (**D**); and, central nervous system (CNS, **E**). **F** shows the sex-specific differences in the adult cohort. *FOXO1*, *BCL2*, and *CREBBP* are predominantly found in the female cohort, while *MYC*, *SMARCA4*, and *DDX3X* are overrepresented in the male cohort (*P* values by one-sided exact Fisher tests, see Supplementary Data 8 for all *P* values and FDRs, *$P \leq 0.05$, **$P \leq 0.01$, ***$P \leq 0.001$.

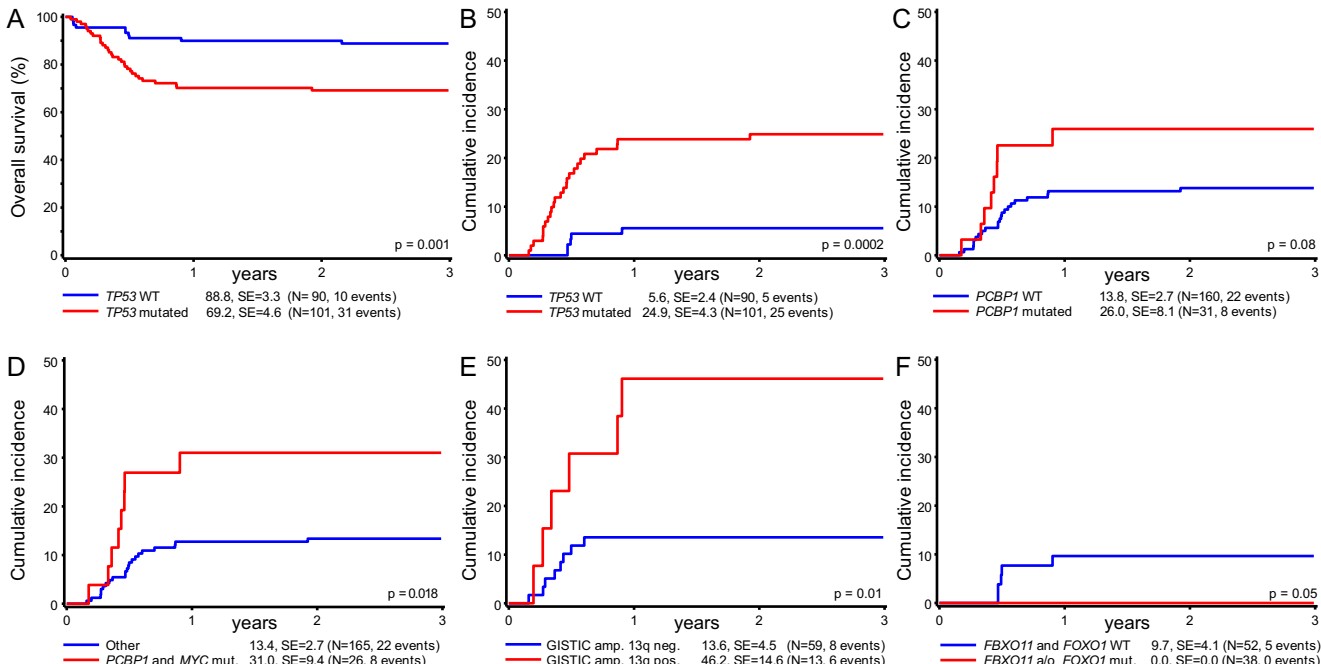

**Fig. 5 Mutations associated with survival and cumulative incidence of relapse in the pediatric cohort. A** The overall survival for pediatric patients with SNV/indels in *TP53* was significantly inferior to patients with *TP53* wild-type (Kaplan–Meier survival estimation, log-rank test). **B** *TP53*[mut] cases also showed an increased risk of relapse (cumulative incidence, Gray's test[20], no adjustments for multiple comparisons in this descriptive context). **C** Mutations in *PCBP1* show a tendency toward prognostic relevance. **D** Cases with lymphomas harboring *MYC* and *PCBP1* mutations are characterized by an increased risk of relapse. Gains or amplifications on 13q31 specific to *GPC5/MIR17HG* were significantly associated with a higher risk of relapse (**E**). The lowest incidence of relapse (0/38 cases) was observed for the *TP53*[wt] sub-group and mutations in *FBXO11/FOXO1* (**F**).

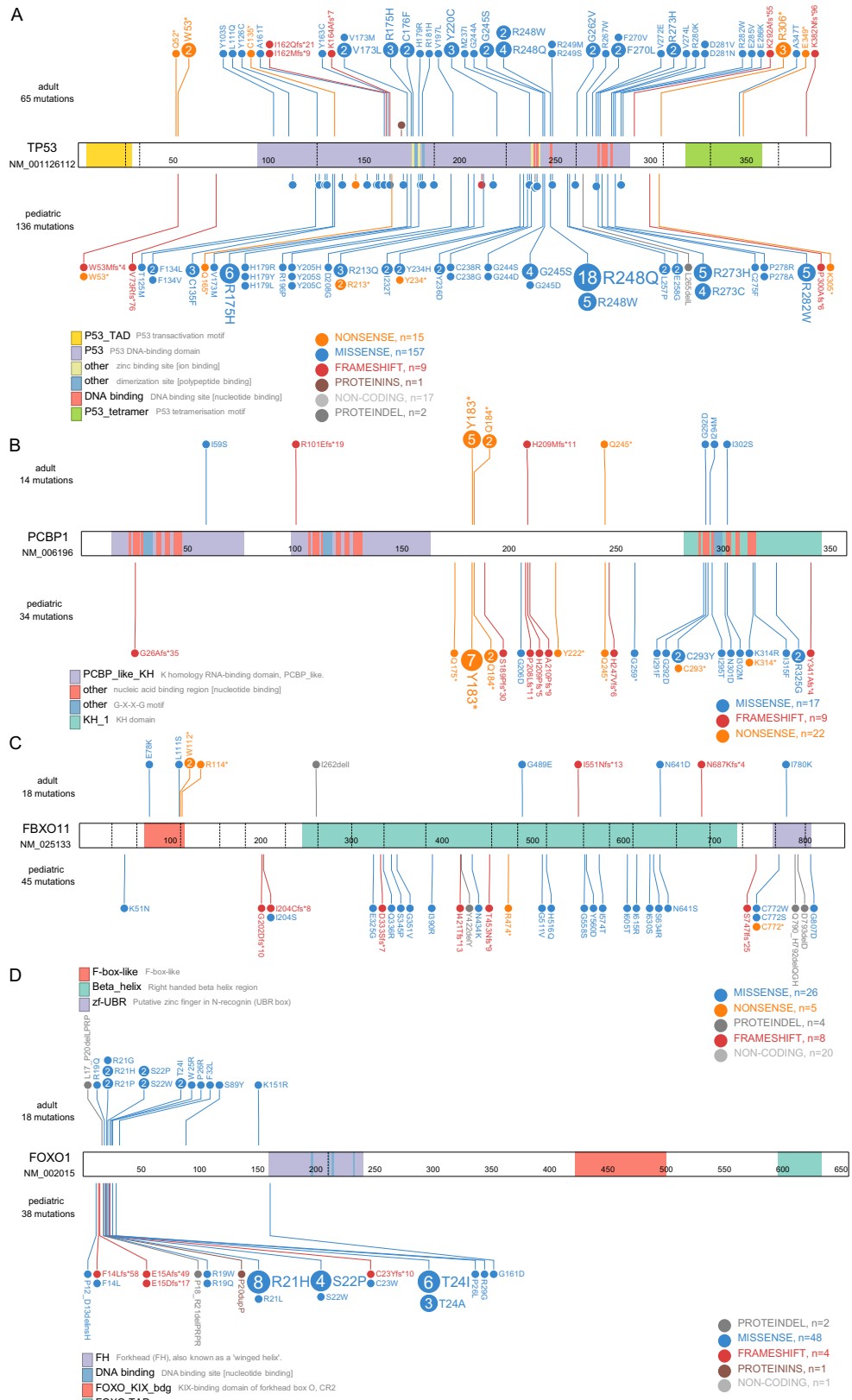

**Fig. 6 Mutational overview of *TP53*, *PCBP1*, *FBXO11*, and *FOXO1* at the amino acid level in the adult (top) and pediatric (bottom) cohort. A** *TP53* shows an accumulation of known hotspots at R175, R248, R273, R282, and G245 as found in other malignancies. **B** *PCBP1* shows a hotspot truncation mutant (Y183*) identified in both age groups. **C** *FBXO11* alterations are unique except for W112*. Most mutations were found in the β-helix domain. **D** In total, 74% of all pediatric *FOXO1* mutations span the Akt phosphorylation motif on R19, R21, and T24. Exon boundaries are indicated using dashed lines.

SNV/indels could also be observed in the rituximab cohort (Supplementary Fig. 7B).

The mutational status of *PCBP1* tended to be associated with an increased incidence of relapse as well (Fig. 5C, $26 \pm 8\%$ vs $14 \pm 3\%$, $P = 0.08$). Interestingly, all patients with *PCBP1* mutation who suffered relapse also showed *MYC* alterations (Fig. 5D, $13 \pm 3\%$ vs $31 \pm 9\%$, $P = 0.018$). As PCBP1 was shown to interact with MYC and regulate its IRES-dependent expression, it is possible that the truncation mutant Y183* of PCBP1 identified in seven out of eight relapse cases might have an impact on *MYC* expression (Fig. 6B and Supplementary Data 9)[31,32].

Very interestingly, we discovered with respect to SCNAs that the highly specific focal gains and amplifications of *GPC5/MIR17HG* were also significantly associated with inferior survival (Fig. 5E, $P = 0.024$). For cases with available SNP measurement, 6/13 cases showing this *GPC5/MIR17HG* lesion relapsed, whereas only 10/59 relapsed without (53% versus 83% relapse-free follow-up).

Of the 13 cases with *GPC5/MIR17HG* lesions, 11 also had a *TP53* mutation. Of these double-hit cases, 5/11 relapsed (55% relapse-free follow-up). In contrast, of the 29 cases in our study with *TP53* mutation but without *GPC5/MIR17HG* copy gain, only 7 relapsed (75% relapse-free follow-up). While these numbers hint to a double-hit BL subtype with a very high relapse probability, this hypothesis requires further validation with larger case numbers.

Notably, we observed a good-risk sub-cohort of 38 samples (20%) without relapses for patients with *TP53*wt status and additional *FOXO1* and/or *FBXO11* mutations ($P = 0.05$, Figs. 5F, 6C, D).

## Discussion

In general, the process of BL lymphomagenesis is thought to depend on three different important pathways that are partially interconnected: escape of apoptosis, deregulated PI3K signaling, and changes in the cell cycle leading to proliferation and survival of cancer cells[33]. In all cases of BL/B-AL pathogenesis, deregulated *MYC* signaling is a main driver. Due to *MYC* mutation, proliferation and apoptosis are uncoupled, consequently proliferation is stimulated and apoptosis abrogated. Deregulation and progression of cell cycle is augmented by several mutations, for example by a mutation in *CCND3*, *TCF3*, *ID3*, and *TP53*[9,26,34,35].

Here, we analyzed a large cohort of pediatric BL, based on high-quality DNA from fresh-frozen samples and an analysis pipeline validated by comprehensive Sanger sequencing, to have a calling sensitivity of 0.993 and specificity of 0.989. We systematically compared the mutational spectrum with adult BL as only limited information on the differences/similarities to pediatric cohorts are available[22,36]. In addition to the well-known MYC driver we identified recurrent SNV/indels in the *ID3-TCF3-CCND3* pathway in 87% of cases as a potential second hit in our pediatric cohort. This is in contrast to the adult cohort, in which alterations of the *ID3-TCF3-CCND3* pathway were identified in only 63%. However, SNV/indels of *BCL2* or *YY1AP1* were nearly exclusively identified in adult BL cases. Our cutoff-free analysis of the mutation landscape over age suggests that the biological transition of mutational profile occurs between 25 and 40 years, i.e., later than the typically assumed adult threshold.

The question of why more male patients are affected by BL remains unanswered. Therefore, we analyzed gender-specific alterations in both cohorts and, strikingly, found a very strong bias of *DDX3X* mutations towards male individuals, which was very recently confirmed by Gong and colleagues[37]. In total, 64% of all SNV/indels identified in the pediatric relapse cohort were nonsense or frameshift variants of *DDX3X*, thought to decrease the expression level of DDX3X.

ARID1A and SMARCA4, two subunits of the SWI/SNF complex, were frequently targeted by mutations in pediatric BL samples. For ARID1A, a functional compensation by ARID1B and ARID2 has recently been shown[38]. Hence, it would be reasonable to analyze these genes or other members of this complex in future studies to determine if they can indeed functionally compensate for the loss of function of ARID1A, as suggested by our mutational analysis. Interestingly, SNV/indels of genes encoding SWI/SNF components result in increased activity of the histone methyltransferase EZH2, which can be targeted by EZH2 inhibitors such as tazemetostat. Tazemetostat shows promising activity in aggressive B-cell lymphoma, such as r/r GCB-type DLBCL, especially when *EZH2* is mutated[39,40]. In addition, loss of the SWI/SNF subunit ARID1A is further reported to be associated with increased sensitivity to PI3K/Akt inhibition in cancer. However, the role of this potential crosstalk mechanism in lymphoma is as yet unclear[41,42].

In DLBCL, SNV/indels of *FOXO1* or *FBXO11* are linked to a higher relapse rate[43,44]. Notably, the pediatric BL sub-group with the most favorable prognosis in this study were patients with SNV/indels in *FBXO11* or *FOXO1* on a *TP53*wt background. The T24I mutation of *FOXO1* identified in pediatric patient samples may lead to an escape of the BCR/PI3K signaling phosphorylation site required for nuclear localization and subsequent expression of target genes[45]. As most SNV/indels are in close proximity to this phosphorylation site, they may have similar effects on the phosphorylation state or transcriptional activity of FOXO1[46]. Taken together, it seems possible that FBXO11, as the direct interaction partner of TP53 responsible for its neddylation and subsequent degradation, and FOXO1 influence survival and proliferation[47]. The least favorable prognosis in this study with a relapse-free follow-up of only 55% was observed for cases double-hit by a *TP53* mutation and simultaneous copy gain or amplification of *GPC5/MIR17HG*. Interestingly, increased expression of *MIR17HG* has already been assigned to inferior outcome in BL[48]. Mechanistically, *MIR17HG* overexpression leads to the down-regulation of PTEN, ultimately resulting in increased PI3K/Akt signaling[49,50].

A key finding of this study is that the prognostic relevance of recurrent SNV/indels affecting *TP53* is in line with our previous clonal evolution data, which point to missing TP53wt protein expression levels in lymphoma samples[11]. The relevance of this finding is underlined by the findings from refs. [35,51], which were published during the revision of this study[36,51]. Both studies show the negative impact of TP53 mutations on survival. The results of these independent studies validate each other and pave the way for rapid translation into upcoming clinical trials by targeting malfunctioning TP53 directly. One strategy might be the inhibition of upstream regulators *MDM2* and *MDM4*, to decrease the ubiquitination and subsequent proteasomal degradation of *TP53*wt. In addition, eprenetapopt, which was recently positively evaluated in a study of patients with MDS, could be a promising drug to restore the TP53wt level[52]. However, the impact of these drugs on mutated TP53 in BL must be verified in vitro and in vivo. In summary, restoring the *TP53*wt level could also decrease *MYC* expression and initiation of apoptosis.

Taken together, we provide a detailed comparison of BL in different age groups that will contribute to a sub-group-specific understanding of BL biology and future treatment options.

## Methods

**Samples and cell lines.** This study complies with all relevant ethical regulations and was approved by the ethics committee of the medical association Westfalen-Lippe, Germany, and the University of Muenster, Germany (pediatric cohort: 2015-495-f-S, adult cohort: 2017-534-f-S). In total, fresh-frozen samples of 298 mature B-cell lymphoma for 191 pediatric BL patients, 97 adult BL patients, and 10

BL cell lines were evaluated (Supplementary Fig. 8). For 86 of the 191 sequenced pediatric BL or Burkitt leukemia (B-AL) cases, additional paired fresh-frozen germline samples were available. All 191 pediatric cases were registered in the NHL-BFM data center and received confirmation of diagnosis by central reference laboratories of the NHL-BFM study group. Pediatric tumor samples were reviewed by expert hematopathologists and classified according to the WHO 2016 guidelines. They were treated according to the consecutive uniform protocols NHL-BFM95, B-NHL BFM04 or NHL-BFM Registry 2012[53]. Clinical data for pediatric patients were obtained from the NHL-BFM study center in Münster, Germany. Molecular features of 18 of the 191 pediatric cases have been described already in previous publications of the ICGC MMML-Seq consortium[8,11]. For the analyses of adult BL, samples were contributed by experienced laboratories for 97 adult BL patients. Written informed consent was obtained from all patients and/or their legal guardians. Patients did not receive any financial compensation for participation.

Cell lines Blue-1, Daudi, Dogkit, Gumbus, Jijoye, Raji, and Ramos were obtained from Louis Staudt (National Cancer Institute, Bethesda). Namalwa and BL-60 were obtained from Stephan Mathas (Max Delbrueck Center, Berlin). BL-70 was obtained from the DSMZ, Braunschweig, Germany (catalog number ACC 233).

**Tissue handling**. DNA was extracted from fresh-frozen tumor tissue, bone marrow, or effusion samples using the DNeasy Blood and Tissue kit (Qiagen, Hilden, Germany). For the isolation of corresponding germline DNA, peripheral blood or bone marrow without blast infiltration was used.

**Targeted deep sequencing and data analysis**. Targeted deep sequencing was performed for 134 genes previously identified to be recurrently mutated in BL/B-AL (Supplementary Table 3). Previous publications and publicly available data from the International Cancer Genome Consortium (www.ICGC.org) were screened for genes with recurring genomic alterations (≥ 2 SNV/indels in the same gene). A Nextera rapid capture custom kit was designed using the Illumina DesignStudio. For every gene, all regions in which previous SNV/indels had been described were covered. For 40 of these genes, the entire coding sequence was covered because previously described alterations were scattered over the entire coding sequence or they were exceptionally frequently affected by SNV/indels according to previous reports. Exact positions of all probes used for enrichment are listed in Supplemental Data 2. Library preparation was performed according to manufacturer recommendations (Illumina, San Diego, USA). Qubit dsDNA assay and Bioanalyzer High Sensitivity DNA chips were used for DNA quantification and library validation. Targeted deep sequencing was performed on a MiSeq-Sequencer using MiSeq Reagent Kit v2 (300 cycle) with 24 samples per run.

An overview of our analysis pipeline with integrated methods and utilized external databases is shown in Supplementary Fig. 1; key steps are described next. All somatic mutations are provided in Supplementary Data 2.

**Sequence alignment**. Measured sequence reads were preprocessed and quality-controlled using cutadapt 1.16, Trim Galore! 0.5.0 and FastQC 0.11.5[54–56]. Trimmed reads were aligned against the current human reference genome from the Genome Reference Consortium (GRCh38) using HISAT2 v2.0.4[57,58].

**Variant discovery, quality control on read and sample level, final cohort sizes**. For variant discovery, we utilized the Genome Analysis Toolkit v4.0.6.0[59] and Mutect2[60]. Only reads aligned by HISAT2 that also passed GATK and Mutect quality control filters were utilized for subsequent variant discovery.

**Basic variant filtering**. To build a panel of normal variants (PON)[61], we performed variant discovery with the same experimental and analytical pipeline for all normal controls. A variant was included in the PON if Mutect determined it as significant in at least two independent subjects. This PON was subsequently used to filter germline variants and potential pipeline-specific artefacts when applying Mutect. For tumor samples for which DNA sequencing of paired normal cell samples was available, we additionally utilized matched normals for a more specific paired statistical variant analysis by Mutect. Otherwise, we used the unpaired analysis mode. In addition, we used the gnomAD database as large population germline resource based on the Exome Aggregation Consortium ExAC[62] for basic filtering.

**Variant annotation and advanced filtering**. Next, we applied an optimized multistage filter hierarchy to reach maximal specificity of somatic mutation calls. All filter steps in the applied order are listed in Supplementary Data 3. For this hierarchy, we annotated discovered variants with their transcript and protein level consequences using TransVar 2.4.0[63] and the NCBI RefSeq gene models[64]. In case of multiple RefSeq transcripts per gene, we annotated each variant with the one leading to the strongest possible biological consequence on protein level according to TransVar. For mutation overview plots, we selected the first principal transcript of the respective gene according to the APPRIS database[65]. In addition, we annotated variants with confirmed somatic mutations according to the Catalogue

Of Somatic Mutations In Cancer (COSMIC v85)[61], the NCBI database of common human variants (>=5%) in any of the five large populations from dbSNP build 151[66], and NCBI ClinVar[67] (version 2018-04) using vcfanno v0.3.0[68].

**Variants called somatic**. Based on variant statistics from Mutect, GATK, and all annotations, our filter hierarchy called 0.25% of all variants in this targeted panel as somatic mutations for the pediatric sub-cohort, i.e., 7.32 on average per sample. For adult BL patients, 0.33% were called somatic (8.49 per sample) whereas for cell lines, 0.52% were called somatic (13.29 per sample). See Supplementary Data 3 for detailed mutation counts and percentages remaining after each filtering step.

**Identification of cancer genes by mutation abundance**. To test if genes depicted in our oncoplots (with > =10% cohort mutation frequency, Fig. 1a, b) were positively selected by mutation processes, we applied the dNdScv tool to all somatic mutations (including synonymous ones)[24]. All oncoplot genes except for *TCF3* had a dN/dS ratio significantly greater than 1 ($q_{dNdS} < =0.1$), suggesting an excess of coding mutations (Supplementary Data 5). The likely reason for the negative result for *TCF3* is alternative transcripts. We discovered 40 missense mutations for NM_001136139 [https://www.ncbi.nlm.nih.gov/nuccore/NM_001136139] (*TCF3* transcript variant 2, protein variant NP_001129611 [https://www.ncbi.nlm.nih.gov/protein/NP_001129611]), whereas only 2/40 were counted by dNdScv, as 38/40 are intronic in context of the default transcript NM_003200 [https://www.ncbi.nlm.nih.gov/nuccore/NM_003200].

**Additional tools and software utilized for sequencing analysis**. For various analysis tasks in the sequencing pipeline, we used bedtools[69], the Integrated Genomics Viewer[70], the Picard toolkit[71], and SAMtools[72]. For analysis pipeline orchestration including parallel remote analysis jobs on high-performance clusters as well as for most visualizations including oncoplots, we used MATLAB® (versions R2018a-R2020a, The MathWorks® Inc., Natick, Massachusetts, USA). R (version 3.6.3, R Foundation for Statistical Computing, Vienna, Austria), Python (version 2.7-3.X, Python Software Foundation, Wilmington, Delaware, USA), and GNU parallel were used for running various tools or for local parallelization[73]. Needle plots of mutation profiles were created using ProteinPaint[74]. All used tools are summarized in Supplementary Table 4.

**Validation by Sanger sequencing**. Validation by Sanger sequencing was performed for a subset of the 191 pediatric patient samples and cell lines; the genes *ID3*, *CCND3*, and *TCF3* were sequenced in 72/191 pediatric patient samples[6]. *TP53* was sequenced in 57 cases, *FOXO1* and *FBXO11* in 25, *PCBP1* in 13, and *P2RY8* in 12/191 cases (for selected primers, see Supplementary Table 6). Sequence analysis was conducted by LGC genomics GmbH, Germany.

We performed a 1:1 comparison with somatic SNV/indels discovered by targeted DNA sequencing. Before validation, 284 variants were called by Sanger sequencing and/or by the DNA-sequencing pipeline in regions covered by Sanger sequencing. 243/284 were matches (identical genomic HGVS from both pipelines). An additional 10 were matches after manual review (e.g., HGVS annotation ambiguities). In all, 26/284 were discovered by both pipelines, but filtered by the DNA-seq pipeline; after manual review, the DNA-sequencing filtering was confirmed in each case. For completeness, in the same regions covered by Sanger sequencing, additional 272 variants were discovered by DNA sequencing, but not called due to the filter hierarchy. Notably, two variants discovered by Sanger sequencing were not rediscovered by targeted DNA sequencing (hard false negatives before filtering). This corresponds to a false negative rate of 0.007 and a sensitivity of 0.993. One missing aberration was a mutation in the cell line Namalwa at 1:g.23559108_23559207delinsCTTTGATGCAACCATGGGCAAG TCACTTGTCCCTCTCTGGCCTCAGTTTCCCTAAACCGAGTGAGTGGCA ATTTTTAAAAACGTCTGCCAACTCCAGGAC, which might be too long for the sequencing technology or Mutect to recognize. The second aberration from an initial tumor sample was a likewise length outlier: 1:g.23559159_23559248delCAG GGGCTGGCTCGGCCAGGACTACCTGCAGGTCGAGAATGTAGTCGATGAC GCGCTGTAGGATTTCCACCTGGCTAAGCTGAGTGCCTC. Finally, three of the 284 somatic SNV/indels called by targeted DNA sequencing were not part of the Sanger validation set. This corresponds to a false positive rate of 0.011 and a specificity of 0.989 for DNA sequencing. However, further manual investigation via visualization in IGV showed that all three were present in the raw targeted sequencing data[70]. For a detailed run-down of this validation of our targeted sequencing measurement and analysis pipeline, including detailed filter reasons for each variant if any, see Supplementary Data 4.

**Analysis of somatic copy number aberrations (SCNAs)**. Single-nucleotide polymorphism arrays were measured by LIFE & BRAIN genomics, Germany, using Human OmniExpress v1.3 (Illumina). Sufficient material for SNP arrays was available for 72 pediatric lymphoma samples and 21 matched germline samples. Raw data were preprocessed according to manufacturer instructions (Illumina Genome Studio v2.0.3)[75]. For allele-specific copy number segmentation at the sample level, we utilized ASCAT v2.4.3. ASCAT also estimated sample ploidy and purity (i.e., the cell fraction originating from aberrant tumor cells, as opposed to

non-aberrant bystander cells). Recurrent SCNAs at the cohort level were identified and statistically evaluated using GISTIC 2.0[76].

**Mutation clonality by integrative analysis of targeted sequencing and SNP results.** To identify potential early mutations in the pathogenesis, we estimated mutation clonality. First, we integrated SNP array results (tumor purity, ploidy, and copy numbers) with variant allele frequencies from the targeted sequencing data to estimate the cancer cell fractions (CCF) that harbor-specific mutations. We utilized the formula:[77]

$$f_{CCF} := \frac{f_{VAF}}{f_{purity}} \cdot \left( \left(1 - f_{purity}\right) \cdot n_{CN,normal} + f_{purity} \cdot n_{CN,tumor} \right) \quad (1)$$

where $f_{VAF}$ denotes variant allele frequency, $f_{purity}$ denotes tumor purity, $n_{CN,normal} = 2$ assuming diploid normal bystander cells, and $n_{CN,tumor}$ denotes tumor cell copy numbers as estimated by ASCAT. Then, we defined clonal variants using the established threshold of $f_{CCF} \geq 0.9$[78]. Estimated Cancer Cell Fractions can be found together with all called mutations in Supplementary Data 2 (see column group Variant def./call quality/details for tumor).

**Statistics and reproducibility.** No statistical method was used to predetermine sample size; all samples of matching diagnosis with enough available sample material were selected for this study. The experiments were not randomized. Several samples were excluded before the final analysis for QC reasons: five samples from the adult cohort due to <10% of the median mapped read count per sample, six samples from the adult cohort due to potentially mismatching disease entity after pathology revision, one pediatric sample due to cell material issues (very low allele frequencies even at SNP loci), three pediatric samples due to potentially wrong channel assignment or mismatching (tumor, normal) pairs, and one pediatric sample due to NHL as second malignancy. The investigators were not blinded to allocation during experiments and outcome assessment.

For binary comparisons between sub cohorts, e.g., pediatric versus adult, and if not otherwise specified, P values were calculated with one-tailed Fisher exact tests. A significance threshold of α = 0.05 was used for all Fisher exact tests (bar graphs). Asterisks indicate significance of single tests as follows: \*$P \leq 0.05$, \*\*$P \leq 0.01$, \*\*\*$P \leq 0.001$. In particular, we tested for associations in gene mutation status with age, comparing pediatric (≤18 years) versus adult (>18) patients. Likewise, we tested for significant differences between female and male patients within each age group, as well as the following clinically defined sub cohorts within pediatric patients: <10 years of age versus ≥10 years; with CNS involvement versus without; with bone marrow involvement versus without; and B-AL versus BL. For each comparison, we tested all genes with an overall mutation frequency of at least 2%. False discovery rates (FDR) for sub-cohort comparisons were computed using the Benjamini and Hochberg method over the set of tested hypotheses (genes with at least 2% cohort mutation frequency; cf. Supplementary Data 8)[79]. For significance, we used the prescribed error threshold of $q = 0.1$.

**Survival analysis.** The probability of EFS and overall survival (OS) was estimated using the Kaplan–Meier method, and compared between subgroups using log-rank tests. EFS was defined as the time from diagnosis to the first event, including relapse, death by any cause, or second malignancy. OS was defined as the time from diagnosis to death by any cause. Cumulative incidence functions for relapse were constructed using the Kalbfleisch and Prentice method and compared with Gray's test[20]. For survival data, we performed explorative hypothesis generation by testing various combinations of discovered genetic lesions. Here, no adjustments were made for multiple comparisons. Hence, the resulting significant associations are descriptive, but we consider validation tests for them worthwhile in future independent BL cohorts. Death in remission and secondary malignancy were considered as competing events. All cumulative incidence estimates are given together with their corresponding standard error (±SE).

**Cutoff-free mutations-over-age analysis.** To analyze the age profile of mutations, we estimated mutation density with the Statistics and Machine Learning Toolbox (Matlab R2020a, ksdensity command). To account for artefacts at interval borders, we used the data reflection method. To assess the statistical significance of mutation enrichment at either old or young ages in a cutoff-free way, we applied one-sided enrichment analysis[80] with 1e5 permutations per tail, separately for each gene by ranking patients by their age and counting mutated cases as signature members.

**Reporting summary.** Further information on research design is available in the Nature Research Reporting Summary linked to this article.

## Data availability

The targeted sequencing data EGAD00001007708 in FASTQ.gz format and SNP data EGAD00010002137 in IDAT format generated in this study have been deposited in the European Genome-phenome Archive (EGA) under study accession EGAS00001005270. These data are available under restricted access for German data privacy laws; access can

be obtained via the associated data access committee EGAC00001002105. EGA access will be granted after a data access treaty has been agreed upon with the law department of the University Hospital Muenster. Restrictions include limiting access to those people named in the agreement for the duration of the named project. Typically, access for universities or public research institutions is granted within one month provided there are no required amendments. The processed somatic mutations and copy number aberrations, as well as clinical metadata, are provided in respective Supplementary Data items. The following public data sources were used in this study: The human reference genome from the Genome Reference Consortium (GRCh38) in its pre-indexed form for alignment with HISAT2 [http://daehwankimlab.github.io/hisat2/download/#h-sapiens], the Catalogue Of Somatic Mutations In Cancer (COSMIC, v85) [https://cancer.sanger.ac.uk/cosmic], the NCBI database of common human variants (based on dbSNP build 151, version 2018-04) [https://www.ncbi.nlm.nih.gov/variation/docs/human_variation_vcf]), NCBI ClinVar (version 2018-04) [https://www.ncbi.nlm.nih.gov/clinvar/], NCBI RefSeq gene models via TransVar (file name hg38.refseq.gff.gz.transvardb, downloaded 20190227), gnomAD/ExAC germline variants as provided in the file af-only-gnomad.hg38.ensemble.vcf.gz of the GATK resource bundle originally accessed via ftp.broadinstitute.org/bundle, but since moved by the Broad Institute to Google cloud bucket; see https://gatk.broadinstitute.org/hc/en-us/articles/360035890811-Resource-bundle for access information, and the principal splice isoforms database (APPRIS, version 2020-01-22) [https://appris.bioinfo.cnio.es/#/downloads]. All remaining data are available within the Article Supplementary Information and Supplementary Datasets.

## Code availability

We used the following software programs and packages (in lexical order): ASCAT 2.4.3 [https://github.com/Crick-CancerGenomics/ascat], bedtools 2.27.1 [https://bedtools.readthedocs.io], cutadapt 1.16 [https://cutadapt.readthedocs.io/en/stable/], dNdScv 0.1.0 (20211202) [https://github.com/im3sanger/dndscv], FastQC 0.11.5 [http://www.bioinformatics.babraham.ac.uk/projects/fastqc], Genome Analysis Toolkit (GATK)/ Mutect 4.0.6.0 [https://github.com/broadinstitute/gatk/releases], GISTIC 2.0 [https://github.com/broadinstitute/gistic2], GNU parallel 20161222 [https://www.gnu.org/software/parallel/], HISAT2 2.0.4 [http://daehwankimlab.github.io/hisat2/download], Illumina Genome Studio 2.0.3 [https://sapac.illumina.com/techniques/microarrays/array-data-analysis-experimental-design/genomestudio.html], Integrated Genomics Viewer 2.5.0-2.8.0 [http://software.broadinstitute.org/software/igv/download], MathWorks MATLAB R2018a-R2020a [https://www.mathworks.com], picard 20180706 [https://broadinstitute.github.io/picard/], Protein Paint web app [https://pecan.stjude.cloud/proteinpaint], Python 2.7 and 3.6 [https://www.python.org], R 3.6.3 [https://www.r-project.org], samtools 1.1 [http://www.htslib.org], TransVar 2.4.0 (20180701) [https://github.com/zwdzwd/transvar], Trim Galore! 0.5.0 [https://github.com/FelixKrueger/TrimGalore], and vcfanno 0.2.9 [https://github.com/brentp/vcfanno/releases]. See Supplementary Fig 1 for a schematic overview of main analyses. Detailed descriptions of our analyses are provided in "Methods" and Supplementary Information. Supplementary Table 5 provides additional details on tool availabilities.

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

## Acknowledgements

We thank Kristian Erdmann, Ulrike Meyer, Jutta Schieferstein, Claudia Sopalla and Patrick Stelmach for excellent technical assistance, the Cells In Motion Cluster of Excellence for the Clinician-Scientist Program (to J.R.), the Mediziner Kolleg of the Medical Faculty of the University of Muenster (to K.R.) and the Deutsche Kinderkrebsstiftung for their support of the NHL-BFM Registry 2012 (DKS 2014.11 and DKS 2016.24 to B.B. and W.W.), which funded parts of this study. Further support was provided by DFG (BU 2453/1-1) and Deutsche José Carreras Leukämiestiftung (R 12/09, 13/2016) to T.B. and by the BRCCH and the Cancer League Switzerland to A.T. We thank all members of the labs for fruitful discussions.

## Author contributions

B.B., J.R., and G.L. designed the study. J.R., T.E., and K.R. performed the research. M.R., A.B., T.B., S.D., A.T., F.F., N.G., T.H., W.K., W.H., W.K., A.R., G.O., I.O., J.Ri., U.K., M.L., B.M., R.S., A.v.S., B.S., and W.W. collected the data or patient/study material. S.S., M.D., M.Zi., M.Za., and M.G. performed statistical and bioinformatic analysis. B.B., U.M., P.B., J.R., K.R., J.F., S.M., M.Zi., M.Za., M.G., and G.L. analyzed and interpreted the data. B.B., U.M., P.B., M.G., and G.L. wrote the manuscript. All authors have read and approved the manuscript.

## Funding

## Competing interests

Torsten Haferlach and Wolfgang Kern declare partial ownership of MLL Münchner Leukämielabor GmbH. The remaining authors declare no competing interests.

## Additional information

[1]Pediatric Hematology, Oncology and BMT, University Hospital Münster, Münster, Germany. [2]Department of Medicine A, Hematology, Oncology, and Pneumology, University Hospital Münster, Münster, Germany. [3]Pediatric Hematology and Oncology, University Hospital Giessen, Giessen, Germany. [4]Department of Pediatric Oncology, Hematology and Clinical Immunology, University Children's Hospital Medical Faculty, Heinrich-Heine-University, Düsseldorf, Germany. [5]Department of Hematology, Oncology and Tumor Immunology, Charité – Universitätsmedizin Berlin, corporate member of Freie Universität Berlin and Humboldt-Universität zu Berlin, Berlin, Germany. [6]Center for Genomic and Computational Biology and Department of Medicine, Duke University, Durham, NC, USA. [7]Pathology, Institute of Medical Genetics and Pathology, University Hospital Basel, Basel, Switzerland. [8]Institute of Medical Informatics, Heidelberg University Hospital, Heidelberg, Germany. [9]Institute of Medical Informatics, University of Münster, Münster, Germany. [10]Institute of Pathology and Neuropathology and Comprehensive Cancer Centre Tübingen, University Hospital Tübingen, Eberhard-Karls-University, Tübingen, Germany. [11]Department of Medicine II, Goethe University, Frankfurt, Germany. [12]MLL Munich Leukemia Laboratory, Munich, Germany. [13]Division of Translational Pathology, Gerhard-Domagk-Institute of Pathology, University Hospital of Münster, Münster, Germany. [14]Department of Pathology, Hematopathology Section, University Hospital Schleswig-Holstein, Kiel, Germany. [15]Section of Pediatric Hematology, Oncology, and Stem Cell Transplantation, Department of Pediatric and Adolescent Medicine, RWTH Aachen University Hospital, Aachen, Germany. [16]Hematology and Oncology, Medical Faculty, University of Augsburg, Augsburg, Germany. [17]Hannover Medical School, Department of Pediatric Hematology and Oncology, Hannover, Germany. [18]Department of Clinical Pathology, Robert-Bosch-Krankenhaus, and Dr. Margarete Fischer-Bosch Institute of Clinical Pharmacology, Stuttgart, Germany. [19]Institute of Pathology, Universität Würzburg and Comprehensive Cancer Centre Mainfranken (CCCMF), Würzburg, Germany. [20]Institute of Human Genetics, Ulm University and Ulm University Medical Center, Ulm, Germany. [21]Department of Pediatric Oncology Hematology, Charité - Universitätsmedizin Berlin, Berlin, Germany. [22]Department of Pediatrics and Adolescent Medicine Division of Pediatric Hematology and Oncology, Medical Center Faculty of Medicine, University of Freiburg, Freiburg im Breisgau, Germany. [23]Pediatric Hematology and Oncology, University Medical Centre Hamburg-Eppendorf, Hamburg, Germany. [24]These authors contributed equally: Birgit Burkhardt, Ulf Michgehl, Jonas Rohde. [25]These authors jointly supervised this work: Michael Grau, Georg Lenz. ✉email: birgit.burkhardt@ukmuenster.de

