## [Peer Review File · Nature Communications]

Clinical relevance of molecular characteristics in Burkitt lymphoma differs according to ageREVIEWER COMMENTS

Reviewer #1 (Remarks to the Author):

Burkhardt et al present a genomic analysis of 191 pediatric and 97 adult Burkitt lymphoma cases. They assess the mutation frequency of 134 genes via targeted sequencing. The authors study the distribution of mutations in 14 genes found to be mutated in at least 10% of samples and test for differences between young and old, male and female and other clinically relevant groups. Interestingly, a number of genes, but not all, differ in mutation prevalence between paediatric and adult cases. Intriguingly, the genetic perspective indicates a divergence of the underlying genomic mechanisms at around the age of 30 rather than adolescence. Lastly the authors describe TP53 mutations as a strong unfavourable biomarker, as well as candidate markers PCBP1 (+MYC) and FBXO11 or FOXO1.

Previous works include whole genome sequencing of 101 cases from malaria endemic and sporadic subtypes (Panea, Blood 2019) or 106 pediatric cases (Grande, Blood 2019). A very recent analysis reveals TP53 mutations to be an unfavourable prognostic marker in 95 paediatric non-Hodgkin lymphomas and explored the effects of different allelic configurations in detail (Newman, Leukemia 2021).

The latter study bites somewhat into the novelty of TP53 as a prognostic marker, but from a clinical point of view it is very useful to see the two studies validating each other.

That said the given study is unique in its comparative nature of paediatric and adult BL cases and will be interesting for the field.

The study is overall well conducted and the manuscript well written. I commend the authors for making all their data available, which will increase the impact of this study.

I have the following comments, which I believe can be readily addressed.

1. As any gene will accumulate somatic mutations throughout life it is important to quantify the excess of coding mutations as a measure of genuine selection. dN/dS is a suitable tool to do this (Martincorena, Cell 2016). I wouldn't expect that to dramatically change the gene list as the authors use a conservative 10% cutoff, but it would strengthen the findings as it would enable the authors to quantify how many of the observed coding changes in each gene are the result of selection. Similarly there is a statement that each case harbours an average of around 7 coding changes, which is true, but would probably be an overestimate of the number of drivers.

2. The authors should ensure that all statistical tests are adjusted for multiple testing. The authors say in the methods section that Benjamini-Hochberg adjustment has been applied, but it's not immediately evident that this was the case for every comparison.

3. Could the authors validate some of the more speculative biomarkers using data from one of the studies mentioned above?

Minor:

p.2 B-AL not defined

p.2 It would be helpful to report the number of genes sequenced

p.3 SNV/indels -- perhaps the term "point mutations" would read better

Figure 3A has no axis label.

Reviewer #2 (Remarks to the Author):

In the present work, Burkhardt et al. describe mutations/indels from the largest cohort of pediatric Burkitt lymphomas so far available in the literature. Of interest, the authors compared mutational profiles of pediatric BL cases with adult BL, identifying genomic alterations specific for the pediatric age spectrum. Moreover, they identified mutations on p53 with negative prognostic impact both for patients treated according to the NHL-BFM treatment protocol and those treated with rituximab. The study is extremely well conducted and all the analyses are very well detailed.

I have only some minor comments for the authors:

1) Line 87: B-AL is mentioned for the first time. The authors should define here the abbreviation, not at line 339. Moreover, in the supplementary material, B-AL cases are indicated as B-ALL, which usually refers to B-cell precursor acute lymphoblastic leukemia, not Burkitt leukemia.

2) At line 343, the authors state that tumor samples were classified based on the WHO 2008. However, at line 88, they state that cases with 11q aberration have been excluded according to the WHO 2016. This seems a bit contradictory. Why do they not refer to the most recent WHO update only?

3) Line 130 and Figure 2: it could be useful to annotate the position of miR-17-92 cluster locus in the figure.

4) At line 357, the authors state that PB/BM samples without blast infiltration were used to isolate germline DNA. I was wondering if the presence of minimal disseminated disease has been excluded by some molecular method. In this regard, it could be important to specify if a VAF cut-off has been applied to call somatic variants (e.g. above 10%? If so, the presence of MDD could not be of relevance)

5) Line 386: please define PON. I understand it means "Panel Of Normal variants", however those who are not familiar with this term could miss the meaning.

6) It is not clear to me which samples were used for validation by Sanger sequencing. At lines 424-425, the authors refer to 91 and 73 samples based on tested genes. Were some patients' tested for both ID3, CCND3, TCF3 and TP53, FBXO11, FOXO1 mutations? Were these patients from the discovery set (191 cases analyzed by WES) or they represent an independent cohort? I think these details are of importance and need to be mentioned when describing the study cohort.

Reviewer #3 (Remarks to the Author):

General comments

The authors aimed to conduct a comparative study in paediatric and adult Burkitt lymphoma and report on both age-associated and prognostic genomic alterations. They include a cohort of 191 paediatric cases and 97 samples from adults in the study which are large cohorts to study this disease and make some meaningful conclusions. The methodology is also appropriate and data analysis is mostly performed well. However, there are some fundamental issues with this manuscript, the results reported and claims by the authors.

- The manuscript is badly structured and is very difficult to follow requiring substantial editing.
- It is hard to understand what patients are included in each analysis.
- Fundamentally the paper lacks novelty as the major finding that they claim are the first to report TP53 as the first molecular marker for relapse incidence in pediatric BL for use in clinical trials is false. This has already been reported by Newman et al in Leukemia <https://www.nature.com/articles/s41375-021-01444-6> who were the first to report this at the 6th International Symposium on CAYA in NHL in October 2018 (<https://onlinelibrary.wiley.com/doi/full/10.1111/bjh.15536> abstract 93).

- The discussion lacks structure and references. Surprisingly, it also includes little discussion about their main claim of TP53 as a prognostic marker.

Specific comments:

The paediatric cohort includes both Burkitt lymphoma and leukemia cases but this data should not be combined. It is not clear how similar the underlying genetics are for the two diseases. The authors could focus on comparing these two groups as this would be a novel study.

The title and abstract are misleading as it is not a pure Burkitt lymphoma cohort and includes leukaemia cases (assuming Burkitt leukemia and not acute lymphoblastic leukemia)

There is confusion on terminology and alternating between B-AL and B-ALL . B-AL is used in the Results and Methods to indicate Burkitt leukemia, but B-ALL is used in Supplementary Table 1 and Supplementary Data 1 (Cohort overview). B-ALL are B-cell acute lymphoblastic leukemia cases and according to WHO Classification of Tumours of Haematopoietic and Lymphoid Tissues, the term B-ALL should not be used to indicate Burkitt leukaemia/lymphoma.

Authors discuss the differences between adult and paediatric cohorts, but do not state if there were genomic differences between the cases of paediatric BL and paediatric B-AL.

Some sentences are 'chatty' or not specific e.g. line 73 – "Only about 80%" and line 75 – " within a few months" should be more specific. More examples provided later.

On Line 56 the authors claim that their findings "will be used in future clinical trials". This is very bold statement based on this manuscript. However, this work, if substantially edited, does validate the findings of Newman et al in leukemia and will help support its inclusion in a clinical trial.

Line 47 – They state there are outcome differences between pediatrics and adults but don't elaborate.

Line 56 – Weird reporting of survival stats, the confidence intervals make it hard to understand.

Line 59 – They should be consistent when describing the disease in children and adults. The authors say "most common" for pediatrics and then use a percentage for adults. They also do not distinguish between Burkitt lymphoma and Burkitt leukemia and justify why they can combine the cases which is essential to believe the results.

Line 81 – "191 paediatric BL" this is very misleading as they are not all Burkitt lymphoma

Line 83 – If GPC5 is prognostically significant then why isn't this mentioned in the abstract?

Line 87 – referred to B-AL without defining the term

Line 87 – Authors refer to the leukemia samples in the text as "B-AL" but cohort table has "B-ALL", is this a typo in the cohort table column?

Line 92 – The description of the adult cohort is very brief

Line 96 – Authors state "1399 alterations" for pediatric cases but "824 SNV/indels" for adults. Need to be consistent when reporting and clarify what is meant by which ever terminology they use.

Line 95 – this section talks about paediatric BL, authors need to clarify if this analysis is only the 103 paediatric BL samples, or the wider cohort containing the additional 88 B-AL/B-ALL samples.

Line 99 – The authors should describe the variant calling prior to the number of aberrations.

Line 99 – "Fresh frozen samples, including 86 matched normals, and various external databases used by our calling pipeline for variant filtering (Supplementary Fig. 8) guarantee high calling sensitivity and specificity (see methods for details)". This is poorly phrased and should justify why this approach guarantees the sensitivity and specificity in the main text.

Line 100 – Supplementary fig. 8 referred to in the text here, this is the first supplementary figure to be referenced in the text. The authors should re-arrange the order of supplementary figures so this is supp fig 1.

Line 102 – Authors state in the text that 243 variants were detected by sanger sequencing. When filtering supplementary data 4 there appears to be 275 variants if Measurement technology = Sanger-sequencing (validation). Authors should clarify this. We also assume that cell lines were to be included in statistic as one of the variants that could not be validated is in a cell line.

Line 104 – “overlong deletions” not sure what the author are referring to here. In the supplementary data 4 files authors state that the variants are likely too long for DNA seq, if this is what they are referring to it needs to be made clearer in the text.

Line 107 – Authors report the most frequently mutated gene to be MYC with 479 SNV/indels in the section titled “Mutational landscape of pediatric BL”. Supplementary figure 2 shows that this is the number of variants found in paed (293 variants) and adult (193 variants) combined. This should be clarified in the text, or the section title should be changed to reflect that this is the comparison of adults and pediatric cases.

Line 145 – The sentence sounds like more of a discussion point.

Line 165 – For clarity authors could use the phrase mutually exclusive to describe 0/91 adult cases had mutations in both ARID1A and SMARCA4 if I have understood the sentence correctly.

Line 172 – double/triple mutated across the whole cohort? Or just one age group?

Line 178 – Unsure why P2RY8 is reported here. While the percentage is slightly higher in adults this is not a significant difference. This is not comparable to the GNA13 mutation frequency differences.

Line 180 – Authors report a 7% mutation frequency in adult cases here. Authors had previously discussed GNA13 mutation frequencies on line 118 but stated that GNA13 alterations were specific to paediatric cases? Line 118 should be rephrased to emphasise that although GNA13 abnormalities were present in < 10% of adult cases, they were not completely absent from this cohort.

Line 185 – it is unclear whether the BL cohort here is referring to paediatric only or pediatric + adult

Line 193 – Authors state they are the first to report SNV/indels in YY1AP1 in BL, however Schmitz et al 2013 paper includes recurrent YY1AP1 variants in Table 1-

<https://www.ncbi.nlm.nih.gov/pmc/articles/PMC3609867/>

Lines 273-5 – These exact sentences are also present on lines 248-251.

Survival analysis – were the children with B-AL treated with the same protocols as the BL cases?

Line 348 – “adult BL samples were contributed by experienced laboratories for 97 BL patients.” This is insufficient information, and the authors need to expand on this and clarify where the samples have come from and confirm the diagnosis. Has any of this data been published elsewhere?

The Gray’s test is not a standard method and should use cox proportional hazards method

The description of the cohorts does not provide sufficient detail regarding pathology review and clinical data, including treatment protocols for all samples.

The discussion lacks structure and does not reflect the main findings of the paper.

The discussion also lacks references e.g.

Introduction and discussion - Deffenbacher et al 2012 should be referenced which is the first comparison between adult and pediatric BL -

<https://ashpublications.org/blood/article/119/16/3757/29906/Molecular-distinctions-between-pediatric-and-adult>

Lines 277-280 – Describes the pathways thought to be important but don’t reference any papers.

Gehringer et al 2020 discuss the importance of PI3K in BL <https://www.nature.com/articles/s41375-019-0628-0> There are many others in this area that should be included.

Lines 306-310 – The differences in DDX3X mutation frequencies between sex has already been described by Gong et al 2021

<https://www.sciencedirect.com/science/article/pii/S1097276521006250?via=ihub>

Line 323 – TP53 already been shown to be prognostic in paediatric BNHL by Newman et al 2021

<https://www.nature.com/articles/s41375-021-01444-6>

Line Schiffman et al 2011 when referring to 13q -
<https://www.ncbi.nlm.nih.gov/pmc/articles/PMC3204171/pdf/nihms-322704.pdf>

Paragraph commencing on line 291 - Zhou et al 2019 when describing FOXO1 in BL -
<https://ashpublications.org/bloodadvances/article/3/14/2118/260101/Sporadic-and-endemic-Burkitt-lymphoma-have>

Figure and Tables

Cohort table – The authors should be consistent with reporting diagnosis. There is no diagnosis entry for any of the adult cases, this should be clarified even if all adult cases are BL and not B-AL.

Cohort table – There is no stage reporting or LDH levels for the adult cases, again this should be included for consistency.

Figure 1 – data should either be divided by Burkitt lymphoma and Burkitt leukaemia to make it clear what disease we are looking at. Also need to show that they are indeed similar at the genome level and can be combined for a prognostic study.

Figure 2 – the genomes seem quite noisy for Burkitt lymphoma. They only include 72 pediatric cases, were they all Burkitt lymphoma?

Figure 5A is labelled as Overall survival (%) on Y-axis, but the figure legend only mentioned event-free survival.

Figure 6 – the plots are difficult to read and text is too small.

Supplementary Table 1 – should be separated by Lymphoma and leukaemia and clinical parameters such as LDH should be included.

Supplementary data 2 – Authors have included sample IDs in the somatic variant table but for ease of the reader an additional column stating which cohort the sample belongs to (BL/B-ALL) should be added, in addition to a column stating whether they are adult or pediatric cases.

Supplementary data 4 – Authors comment on the number of sanger variants also detected by NGS, but fail to state the number of variants filtered out by their pipeline. Authors should also state in the text the number of variants called by their NGS pipeline that were not present/failed sanger.

Supplementary figure 6B – authors report importance of TP53 in patients treated with Rituximab but there are <5 events. Needs to be shown in a larger cohort before they can conclude this.

Some specific examples of grammatical errors.

Line 47 - "survival improved" are they missing a word? "Survival has improved"

Line 60 – "while it accounts for 1%". What is 'it'. The sentence should be phrased "in comparison BL accounts for ~1%" is a preferable phrase.

Line 67 – "recent reports have contributed to the molecular understanding of BL biology". This is very vague and the authors need to elaborate on this.

Line 67 – The sentences beginning "In diffuse" is poorly worded and they don't introduce why DLBCL is relevant in a BL paper.

Line 79 – "Molecular measurements" do they mean to say "Molecular markers"?

Line 90 – "Roughly 10%" is not very scientific. Suggest "Approximately 10%".

In the term 'p value' the p should be in italics.

REVIEWER COMMENTS

Reviewer #1 (Remarks to the Author):

Burkhardt et al present a genomic analysis of 191 pediatric and 97 adult Burkitt lymphoma cases. They assess the mutation frequency of 134 genes via targeted sequencing. The authors study the distribution of mutations in 14 genes found to be mutated in at least 10% of samples and test for differences between young and old, male and female and other clinically relevant groups. Interestingly, a number of genes, but not all, differ in mutation prevalence between paediatric and adult cases. Intriguingly, the genetic perspective indicates a divergence of the underlying genomic mechanisms at around the age of 30 rather than adolescence. Lastly the authors describe TP53 mutations as a strong unfavourable biomarker, as well as candidate markers PCBP1 (+MYC) and FBXO11 or FOXO1. Previous works include whole genome sequencing of 101 cases from malaria endemic and sporadic subtypes (Panea, Blood 2019) or 106 pediatric cases (Grande, Blood 2019). A very recent analysis reveals TP53 mutations to be an unfavourable prognostic marker in 95 paediatric non-Hodgkin lymphomas and explored the effects of different allelic configurations in detail (Newman, Leukemia 2021). The latter study bites somewhat into the novelty of TP53 as a prognostic marker, but from a clinical point of view it is very useful to see the two studies validating each other.

That said the given study is unique in its comparative nature of paediatric and adult BL cases and will be interesting for the field.

The study is overall well conducted and the manuscript well written. I commend the authors for making all their data available, which will increase the impact of this study.

I have the following comments, which I believe can be readily addressed.

1. As any gene will accumulate somatic mutations throughout life it is important to quantify the excess of coding mutations as a measure of genuine selection. dN/dS is a suitable tool to do this (Martincorena, Cell 2016). I wouldn't expect that to dramatically change the gene list as the authors use a conservative 10% cutoff, but it would strengthen the findings as it would enable the authors to quantify how many of the observed coding changes in each gene are the result of selection. Similarly, there is a statement that each case harbours an average of around 7 coding changes, which is true, but would probably be an overestimate of the number of drivers.

We thank the reviewer directing us to this very useful tool. We applied it to all somatic mutations (including synonymous ones) to test whether the genes depicted in the oncoplot in Figure 1 ($\geq 10\%$ cohort mutation frequency) were positively selected. Using a significance cutoff at $q_{dNdS}=0.1$, all oncoplot genes except for *TCF3* showed a dN/dS ratio greater than 1, indicating excess coding mutations. Analysis results are provided in tabular form in the new Supplementary Data 5. Upon detailed investigation of *TCF3*, we detected that $dNdScv$ only counted 2/40 discovered somatic mutations. The likely reason is that we annotate each somatic mutation with the transcript leading to the strongest consequence on protein level. In case of *TCF3*, we discovered 40 missense mutations in context of the transcript NM_001136139 (Homo sapiens transcription factor 3 (TCF3), transcript variant 2). In context of the default transcript NM_003200 for *TCF3*, 38 of the same 40 variants are intronic. Thus, we clarified in the text (line 128-129) that the detected *TCF3* mutations correspond to the protein variant NP_001129611. This has been described in the revised manuscript (lines 433-441).

2. The authors should ensure that all statistical tests are adjusted for multiple testing. The authors say in the methods section that Benjamini-Hochberg adjustment has been applied, but it's not immediately evident that this was the case for every comparison.

We thank the reviewer to pointing this out. We have not applied the Benjamini-Hochberg adjustment for every analysis. Thus, we have clarified our approach in the revised manuscript after line 509. We did not apply alpha correction for the analysis of survival data since we were interested not only in a predefined set of comparisons but wanted to look for interesting combinations of genes looking at the data. All tests were descriptive and explorative. However, the Benjamini-Hochberg multiple hypothesis

correction has been systematically applied for each subcohort comparison analysis over the set of tested hypotheses (i.e., genes with at least 2% cohort mutation frequency; Supplementary Data 8).

3. Could the authors validate some of the more speculative biomarkers using data from one of the studies mentioned above?

We thank the reviewer for this valid suggestion. All of our top-ranked (>10% VAF) genes were also found in the studies mentioned above¹⁻³. While we agree with the reviewer that comparing all different datasets in detail would be desirable, this is unfortunately out of the scope of the current manuscript and should be approached in a future research project.

Minor:

p.2 B-AL not defined

We thank the reviewer and accordingly we have defined the abbreviation in the revised manuscript in line 60: "Burkitt lymphoma (BL) and its leukemic manifestation Burkitt leukemia (B-AL)..."

p.2 It would be helpful to report the number of genes sequenced

In our study, 134 genes were sequenced with a targeted approach (see Supplementary Table 3). Additionally, somatic copy number aberrations were analyzed. For clarification we added this information to the introduction line 83 of the revised manuscript. Details can be found in the methods section lines 378-394.

p.3 SNV/indels -- perhaps the term "point mutations" would read better

We thank the reviewer for raising this point and agree, that the term "point mutation" would increase readability. Still, we feel, that the term SNV/indel describes the various genetic alterations identified in this study more precisely. As we not only found single base pair changes but also deletions or insertions of multiple bases and copy number aberrations.

Figure 3A has no axis label.

We thank the reviewer and added the missing axis label.

Reviewer #2 (Remarks to the Author):

In the present work, Burkhardt et al. describe mutations/indels from the largest cohort of pediatric Burkitt lymphomas so far available in the literature. Of interest, the authors compared mutational profiles of pediatric BL cases with adult BL, identifying genomic alterations specific for the pediatric age spectrum. Moreover, they identified mutations on p53 with negative prognostic impact both for patients treated according to the NHL-BFM treatment protocol and those treated with rituximab. The study is extremely well conducted and all the analyses are very well detailed.

I have only some minor comments for the authors:

1) Line 87: B-AL is mention for the first time. The authors should define here the abbreviation, not at line 339. Moreover, in the supplementary material, B-AL cases are indicated as B-ALL, which usually refers to B-cell precursor acute lymphoblastic leukemia, not Burkitt leukemia.

We thank the reviewer for this comment. We have defined the abbreviation in the revised manuscript line 60: "Burkitt lymphoma (BL) and its leukemic manifestation Burkitt leukemia (B-AL)..."

In the revised supplementary material, we corrected "B-ALL" to B-AL" to prevent confusions.

2) At line 343, the authors state that tumor samples were classified based on the WHO 2008. However, at line 88, they state that cases with 11q aberration have been excluded according to the WHO 2016. This seems a bit contradictory. Why do they not refer to the most recent WHO update only?

All cases were classified according to the most recent version of the WHO classification at the time of diagnosis. In order to perform the project in a homogeneous cohort of Burkitt samples, we decided not to include cases with 11q aberration. All cases fulfill the diagnostic criteria of the most recent WHO classification. The reviewer is correct with the proposed clarification that all cases were classified as Burkitt lymphoma according to the recent WHO classification. Please find the clarification in the revised version of the manuscript line 363.

3) Line

130 and Figure 2: it could be useful to annotate of position of miR-17-92 cluster locus in the figure.

We have amended the labeling of Figure 2 by adding the miR-17-92 cluster locus as advised by the reviewer. At the resolution of this figure, the new label appears at the identical position as *GPC5*. Therefore, we have also added precise coordinates and names of affected non-coding genes in the manuscript (lines 143-148).

4) At line 357, the authors state that PB/BM samples without blast infiltration were used to isolate germline DNA. I was wondering if the presence of minimal disseminated disease has been excluded by some molecular method. In this regard, it could be important to specify if a VAF cut-off has been applied to call somatic variants (e.g. above 10%? If so, the presence of MDD could not be of relevance).

The reviewer raises an important issue. For all germline samples central cytomorphological review was performed to rule out the microscopic presence of residual blasts. To confidently exclude any impact of molecular MDD on the results, our variant filtering hierarchy utilizes a conservative cutoff at 10% variant allele frequency for calling somatic mutations (Supplementary Data 3). Thus, we found it dispensable to test for MDD on a molecular level.

5) Line 386: please define PON. I understand it means "Panel Of Normal variants", however those who are not familiar with this term could miss the meaning.

We thank the reviewer for this suggestion and, accordingly, we have now defined the PON abbreviation at line 405 of the revised manuscript.

6) it is not clear to me which samples were used for validation by Sanger sequencing. At lines 424-425, the authors refer to 91 and 73 samples based on tested genes. Were some patients' tested for both ID3, CCND3, TCF3 and TP53, FBXO11, FOXO1 mutations? Were these patients from the discovery set (191 cases analyzed by WES) or they represent an independent cohort? I think these details are of importance and need to be mentioned when describing the study cohort.

The reviewer raises an important point and accordingly we have changed this in text lines 455-456 of the revised manuscript: Sanger validation was performed for a subset of the 191 pediatric cases and cell lines. Not all mentioned genes were measured for all Sanger-sequenced cases. Full details including all sample IDs are provided in Supplementary Data 4.

Reviewer #3 (Remarks to the Author):

General comments

The authors aimed to conduct a comparative study in paediatric and adult Burkitt lymphoma and report on both age associated and prognostic genomic alterations. They include a cohort of 191 paediatric cases and 97 samples from adults in the study which are large cohorts to study this disease and make

some meaningful conclusions. The methodology is also appropriate and data analysis is mostly performed well. However, there are some fundamental issues with this manuscript, the results reported and claims by the authors.

- The manuscript is badly structured and is very difficult to follow requiring substantial editing.

To address the criticism of the reviewer we have restructured the manuscript. Especially among others we have changed the numbering of the supplemental figures and supplemental data files to improve the structure and to increase readability.

- It is hard to understand what patients are included in each analysis.

The reviewer raises an important point and accordingly we have clarified for each analysis how many samples were investigated in these experiments. Details are provided in our replies for each individual point below.

- Fundamentally the paper lacks novelty as the major finding that they claim are the first to report TP53 as the first molecular marker for relapse incidence in pediatric BL for use in clinical trials is false. This has already been reported by Newman et al in Leukemia <https://www.nature.com/articles/s41375-021-01444-6> who were the first to report this at the 6th International Symposium on CAYA in NHL in October 2018 (<https://onlinelibrary.wiley.com/doi/full/10.1111/bjh.15536> abstract 93).

We thank the reviewer for raising this point. In our manuscript we were able to show that the mutational status of TP53 is statistically significantly associated with the incidence of relapse in 191 pediatric patients with Burkitt lymphoma/leukemia. All patients were uniformly diagnosed, staged and treated according to NHL-BFM protocols. We feel that these results are novel and highly relevant for future treatment protocols. The relevance of the finding is underlined by the parallel research project conducted by our colleagues in the United Kingdom and published during the ongoing review process of our manuscript. Newman and colleagues describe a similar finding regarding the prognostic impact of TP53 in a cohort of 64 Burkitt lymphoma patients recruited over a period of 20 years who were treated according to a different treatment regimen. The results of the two independent projects validate each other and allow the rapid translation into upcoming clinical trials that are in preparation. We have discussed this in the revised manuscript on lines 342-346.

- The discussion lacks structure and references. Surprisingly, it also includes little discussion about their main claim of TP53 as a prognostic marker.

To increase readability and to improve the structure of the text, we modified the section on TP53 mutations in the "Discussion" of the revised manuscript (lines 288-354 of the revised manuscript). Among others we have now included the study of Newman et al. as central reference pointing out the need to analyze the TP53 mutational status in patients with Burkitt lymphoma. Additionally, we have updated the respective literature in the "Discussion".

Specific comments:

The paediatric cohort includes both Burkitt lymphoma and leukemia cases but this data should not be combined. It is not clear how similar the underlying genetics are for the two diseases. The authors could focus on comparing these two groups as this would be a novel study.

We thank the reviewer for this suggestion. These data were already included in the manuscript files (see subcohort comparison, now Supplementary Data 8, and Figure 4D). As can be seen in this analysis, CCND3 mutations are detected more frequently in the B-AL cohort (n=88), while GNAI2, RFX7, P2RY8, DDX3X and GNA13 aberrations are predominantly found in the BL cohort (n=99). To additionally illuminate these differences, we have divided the oncoplot for paediatric cases (Figure 1A) into two parts, one representing BL (A) and the other representing B-AL cases (B):

Notably, the current WHO classification and recent manuscripts on Burkitt lymphoma do not discriminate between cases with less than 25% blast count in the bone marrow (BL) and those with blast counts in analyzed samples exceeding 25%^{1,4-11}. In addition, the percentage of bone marrow infiltration can vary according to site of aspirate within a patient and for the method of evaluation (e.g., cytomorphology, flow analysis, immunohistochemistry of bone marrow biopsies etc.). This underlines that the level of 25% blast count in the bone marrow that has been used in the past is arbitrary and does not reflect a biological marker to discriminate two different diseases. This is further supported by the fact that current treatment protocols do not discriminate between Burkitt lymphoma and Burkitt leukemia but recruit both as manifestations of one disease entity. Therefore, we present the pediatric cohort in Figure 1A by combining both BL and B-AL and discuss the differences in Figure 4D.

The title and abstract are misleading as it is not a pure Burkitt lymphoma cohort and includes leukaemia cases (assuming Burkitt leukemia and not acute lymphoblastic leukemia)

There is confusion on terminology and alternating between B-AL and B-ALL. B-AL is used in the Results and Methods to indicate Burkitt leukemia, but B-ALL is used in Supplementary Table 1 and Supplementary Data 1 (Cohort overview). B-ALL are B-cell acute lymphoblastic leukemia cases and according to WHO Classification of Tumours of Haematopoietic and Lymphoid Tissues, the term B-ALL should not be used to indicate Burkitt leukaemia/lymphoma.

We thank the reviewer for pointing this out. We apologize for this point of confusion by using the “B-ALL” abbreviation in supplementary tables/data. Accordingly, we have corrected this to “B-AL” throughout the revised manuscript.

Authors discuss the differences between adult and paediatric cohorts, but do not state if there were genomic differences between the cases of paediatric BL and paediatric B-AL.

We thank the reviewer for this suggestion. These data are presented in the manuscript files (see subcohort comparison, now Supplementary Data 8, and Figure 4D).

Some sentences are 'chatty' or not specific e.g. line 73 – “Only about 80%” and line 75 – “within a few months” should be more specific. More examples provided later.

According to the suggestion of the reviewer we have clarified in the revised version of the manuscript that overall survival (OS) is 80% for adult patients with Burkitt lymphoma (line 75) and that relapses occur shortly after the end of treatment (line 77).

On Line 56 the authors claim that their findings “will be used in future clinical trials”. This is very bold statement based on this manuscript. However, this work, if substantially edited, does validate the findings of Newman et al in leukemia and will help support its inclusion in a clinical trial.

We agree with the reviewer that the independent projects by the UK group and the NHL-BFM group validate each other. Currently the NHL-BFM and the Scandinavian NOPHO group conduct the ongoing trial B-NHL 2013 (NCT03206671). The first author of our manuscript, Birgit Burkhardt, is the international PI of this trial. The results of the current project and the recently published data of the UK group were discussed in the steering committee of the NHL-BFM and NOPHO cooperation. We agreed that the data are highly relevant and aim to translate the results into the stratification and treatment plan of the subsequent trial that we are about to design in the next months.

Line 47 – They state there are outcome differences between pediatrics and adults but don't elaborate.

Given the restrictions for the word count in the abstract (lines 46-58) we focused on the fact that there are differences in outcome between pediatric and adult Burkitt patient cohorts. More details and references are provided in the “Introduction” of the revised manuscript (lines 59-89).

Line 56 – Weird reporting of survival stats, the confidence intervals make it hard to understand.

There are no confidence intervals. For clarity we have added in line 528-530 of the revised manuscript that all cumulative incidence estimates are given together with their corresponding standard error (\pm SE).

Line 59 – They should be consistent when describing the disease in children and adults. The authors say “most common” for pediatrics and then use a percentage for adults. They also do not distinguish between Burkitt lymphoma and Burkitt leukemia and justify why they can combine the cases which is essential to believe the results.

The information about the relevance of Burkitt lymphoma/leukemia in children and adolescents is described in more detail in the revised manuscript line 60 “Burkitt lymphoma (BL) and its leukemic manifestation Burkitt leukemia (B-AL) is the most common subtype of Non-Hodgkin Lymphoma (NHL) in children and adolescents accounting for 48% of NHL”. The source providing the numbers has been cited (reference 2).

As stated above in detail, the current WHO classification and recent manuscripts on Burkitt lymphoma do not discriminate between cases with less than 25% blast count in the bone marrow (BL) and those with blast count in analyzed samples exceeding 25%^{1,4-11}.

Line 81 – “191 paediatric BL” this is very misleading as they are not all Burkitt lymphoma

We thank the reviewer for pointing this out. Accordingly, we have corrected this part and now state BL/B-AL. This has been clarified consistently throughout the revised manuscript.

Line 83 – If GPC5 is prognostically significant then why isn't this mentioned in the abstract?

GPC5/miR17-92 are directly adjacent (chr13:91486342-91520022/chr13:91347820..91354575), as described in more detail in the revised manuscript (lines 143-148). In Figure 6, these loci cannot be resolved individually due to resolution. We have changed the figure text accordingly and, consistently, use "GPC5/miR17HG" for this SCNA in the text. As briefly discussed (lines 335-339), the gain of these non-coding micro-RNAs might be prognostically relevant, not the adjacent coding gene GPC5, which is why we did not include it in the abstract.

Line 87 – referred to B-AL without defining the term

We have defined the abbreviation in the revised manuscript in line 60: "Burkitt lymphoma (BL) and its leukemic manifestation Burkitt leukemia (B-AL)..." The abbreviation has been clarified throughout the revised manuscript.

Line 87 – Authors refer to the leukemia samples in the text as "B-AL" but cohort table has "B-ALL", is this a typo in the cohort table column?

Thank you for this comment about the labeling in the Supplement. In the revised version it has been corrected from B-ALL to B-AL.

Line 92 – The description of the adult cohort is very brief

We agree with reviewer. The samples comprising the adult cohort were obtained from different sources. They were collected retrospectively from the Department of Pathology of the University Hospital Münster, the Institute of Pathology of the University of Würzburg and the Comprehensive Cancer Centre Mainfranken (CCCMF), the Center for Genomic and Computational Biology and Department of Medicine of the Duke University, the Department of Clinical Pathology of the Robert-Bosch-Krankenhaus and Dr. Margarete Fischer-Bosch Institute of Clinical Pharmacology in Stuttgart, the Department for Hematology, Oncology and Tumor Immunology of the Charité Berlin, the Institute of Medical Genetics and Pathology of the University Hospital Basel, the Institute of Pathology and Neuropathology and Comprehensive Cancer Centre of the University Hospital Tübingen, and the MLL Munich Leukemia Laboratory. Due to the nature of the sample collection clinical data were not available for the whole cohort. Therefore, corresponding analyses for this cohort are not possible. We added sample source information for both cohorts in the new Supplementary Table 2.

Line 96 – Authors state "1399 alterations" for pediatric cases but "824 SNV/indels" for adults. Need to be consistent when reporting and clarify what is meant by which ever terminology they use.

We agree with the reviewer that this should be reported using unambiguous terminology and accordingly we have corrected this to "1399 SNV/indels" (line 105).

Line 95 – this section talks about paediatric BL, authors need to clarify if this analysis is only the 103 paediatric BL samples, or the wider cohort containing the additional 88 B-AL/B-ALL samples.

In this line we are referring to the wider cohort of 191 cases (line 103 in the revised manuscript) and accordingly have clarified this.

Line 99 – The authors should describe the variant calling prior to the number of aberrations.

As suggested by the reviewer, we have changed the order. In the results section of the revised manuscript, we first describe our analysis pipeline for variant calling before we report the calling results.

Line 99 – “Fresh frozen samples, including 86 matched normals, and various external databases used by our calling pipeline for variant filtering (Supplementary Fig. 8) guarantee high calling sensitivity and specificity (see methods for details)”. This is poorly phrased and should justify why this approach guarantees the sensitivity and specificity in the main text.

We agree with the reviewer. To address this with your previous comment, we have rewritten the start of the “Results” section “Mutational landscape of pediatric BL” accordingly.

Line 100 – Supplementary fig. 8 referred to in the text here, this is the first supplementary figure to be referenced in the text. The authors should re-arrange the order of supplementary figures so this is supp fig 1.

We thank the reviewer for this advice and restructured the complete file order. Now the figures and tables are introduced chronologically.

Line 102 – Authors state in the text that 243 variants were detected by sanger sequencing. When filtering supplementary data 4 there appears to be 275 variants if Measurement technology = Sanger-sequencing (validation). Authors should clarify this. We also assume that cell lines were to be included in statistic as one of the variants that could not be validated is in a cell line.

The 243 variants counted only the matches between both sequencing technologies after filtering. As we did not want to restrict the validation only to patient samples, the 275 variants detected by Sanger sequencing in Supplementary Data 4 contain variants also found in cell line samples. We agree that these different counts might have been confusing. Therefore, we have now detailed and clarified this in the revised manuscript in the section “Validation by Sanger sequencing” (lines 452-481) as well as in the tabular validation overview (Supplementary Data 4, third worksheet).

Line 104 – “overlong deletions” not sure what the author are referring to here. In the supplementary data 4 files authors state that the variants are likely too long for DNA seq, if this is what they are referring to it needs to be made clearer in the text.

We thank the reviewer for pointing this out and accordingly have clarified in the text that we, indeed, are referring to two deletions that were likely too long for DNA-sequencing. Their precise definitions in HGVS annotation are provided in the methods section for the validation analysis (lines 459-466).

Line 107 – Authors report the most frequently mutated gene to be MYC with 479 SNV/indels in the section titled “Mutational landscape of pediatric BL”. Supplementary figure 2 shows that this is the number of variants found in paed (293 variants) and adult (193 variants) combined. This should be clarified in the text, or the section title should be changed to reflect that this is the comparison of adults and pediatric cases.

As suggested by the reviewer, we have clarified in the text of the revised manuscript (line 114) that the 479 MYC mutations correspond to the full cohort (pediatric and adult cases combined).

Line 145 – The sentence sounds like more of a discussion point.

We agree with the reviewer. As these data are well known, we did not include this in the “Discussion”.

Line 165 – For clarity authors could use the phrase mutually exclusive to describe 0/91 adult cases had mutations in both ARID1A and SMARCA4 if I have understood the sentence correctly.

We agree with the reviewer that “mutually exclusive” is a more concise summary of the finding depicted by the Venn diagram in Supplementary Figure 4b and have amended the text accordingly in line 181 of the revised manuscript.

Line 172 – double/triple mutated across the whole cohort? Or just one age group?

We have clarified this in the revised manuscript and have inserted “of the complete cohort” (line 187).

Line 178 – Unsure why P2RY8 is reported here. While the percentage is slightly higher in adults this is not a significant difference. This is not comparable to the GNA13 mutation frequency differences.

We reported this finding as P2RY8 and GNA13 are thought to act in the same GPCR pathway and can potentially compensate each other. However, we agree with the reviewer that there is no significant difference in the frequency and therefore we changed the section accordingly on lines 193-206 of the revised manuscript.

Line 180 – Authors report a 7% mutation frequency in adult cases here. Authors had previously discussed GNA13 mutation frequencies on line 118 but stated that GNA13 alterations were specific to paediatric cases? Line 118 should be rephrased to emphasise that although GNA13 abnormalities were present in < 10% of adult cases, they were not completely absent from this cohort.

We thank the reviewer for this advice. We changed line 131 accordingly.

Line 185 – it is unclear whether the BL cohort here is referring to paediatric only or pediatric + adult

We thank the reviewer for the comment. Accordingly, we have clarified the text in the revised manuscript (line 200). We updated the text as we found both genes mutated in both cohorts. We also added a link to Supplementary Figures 6B, C where this information can be found.

Line 193 – Authors state they are the first to report SNV/indels in YY1AP1 in BL, however Schmitz et al 2013 paper includes recurrent YY1AP1 variants in Table 1- <https://www.ncbi.nlm.nih.gov/pmc/articles/PMC3609867/>

We agree to the reviewer. *YY1AP1* mutations were identified in our cohort by targeted sequencing. Since we used a targeted approach focusing on genes that had previously been described to be recurrently mutated in Burkitt lymphoma, this part of our study was not capable of reporting any new alterations. The text has been adapted accordingly on page 6 of the revised manuscript.

Lines 273-5 – These exact sentences are also present on lines 248-251.

Survival analysis – were the children with B-AL treated with the same protocols as the BL cases?

We thank the reviewer for pointing this out. The text in lines 273-275 was deleted in the revised manuscript.

Line 348 – “adult BL samples were contributed by experienced laboratories for 97 BL patients.” This is insufficient information, and the authors need to expand on this and clarify where the samples have come from and confirm the diagnosis. Has any of this data been published elsewhere?

The samples comprising the adult cohort were obtained from different sources. They were collected retrospectively from the Department of Pathology of the University Hospital Münster, the Institute of Pathology of the University of Würzburg and the Comprehensive Cancer Centre Mainfranken (CCCMF), the Center for Genomic and Computational Biology and Department of Medicine of the Duke University, the Department of Clinical Pathology of the Robert-Bosch-Krankenhaus and Dr. Margarete Fischer-Bosch

Institute of Clinical Pharmacology in Stuttgart, the Department for Hematology, Oncology and Tumor Immunology of the Charité Berlin, the Institute of Medical Genetics and Pathology of the University Hospital Basel, the Institute of Pathology and Neuropathology and Comprehensive Cancer Centre of the University Hospital Tübingen, and the MLL Munich Leukemia Laboratory. The respective diagnoses were confirmed by local pathologists. A central pathology review is not available. Due to the nature of the sample collection clinical data were not available for the whole cohort. Therefore, corresponding analyses for this cohort are not possible. These data have not been published elsewhere.

The Gray's test is not a standard method and should use cox proportional hazards method.

We used the Gray's test as it is the standard test for comparing cumulative incidence functions. Cumulative incidence functions for relapse are necessary because there are competing events, mainly toxic death. We were interested in relapse rates because these rates can possibly be influenced by risk adapted treatment. Standard cox regression does not account for competing events.

The description of the cohorts does not provide sufficient detail regarding pathology review and clinical data, including treatment protocols for all samples.

We agree with reviewer that this is true for the adult cohort. As outlined above the samples comprising the adult cohort were obtained from different sources. Due to the nature of the sample collection clinical data were not available for the whole cohort. Therefore, corresponding analyses for this cohort are not possible.

The discussion lacks structure and does not reflect the main findings of the paper.

To address the criticism of the reviewer, we have modified the "Discussion". Especially we have discussed in more detail our finding on the prognostic value of *TP53* mutations.

The discussion also lacks references e.g. Introduction and discussion - Deffenbacher et al 2012 should be referenced which is the first comparison between adult and pediatric BL –

We thank the author for this advice and have inserted this reference in the "Introduction" and "Discussion" of the revised manuscript.

Lines 277-280 – Describes the pathways thought to be important but don't reference any papers. Gehringer et al 2020 discuss the importance of PI3K in BL <https://www.nature.com/articles/s41375-019-0628-0> There are many others in this area that should be included.

We agree with the reviewer and accordingly we have added this excellent reference in the "Discussion".

Lines 306-310 – The differences in DDX3X mutation frequencies between sex has already been described by Gong et al 2021 <https://www.sciencedirect.com/science/article/pii/S1097276521006250?via=ihub>

We agree with the reviewer. The study by Gong and colleagues was published after the submission of the current manuscript. We have now added the reference as suggested (lines 310-311).

Line 323 – TP53 already been shown to be prognostic in paediatric BNHL by Newman et al 2021 <https://www.nature.com/articles/s41375-021-01444-6>

As outlined above, the study by Newman and colleagues was published during the revision process of the current manuscript. To address the important point of the reviewer we have discussed the study and its findings in the revised "Discussion" (lines 342-346 of the revised manuscript).

Line Schiffman et al 2011 when referring to 13q -
<https://www.ncbi.nlm.nih.gov/pmc/articles/PMC3204171/pdf/nihms-322704.pdf>

As suggested, we have included this reference.

Paragraph commencing on line 291 - Zhou et al 2019 when describing FOXO1 in BL –
<https://ashpublications.org/bloodadvances/article/3/14/2118/260101/Sporadic-and-endemic-Burkitt-lymphoma-have>

As suggested, we have included this reference.

Figure and Tables

Cohort table – The authors should be consistent with reporting diagnosis. There is no diagnosis entry for any of the adult cases, this should be clarified even if all adult cases are BL and not B-AL.

As stated above in detail, the samples comprising the adult cohort were obtained from different sources. The diagnosis was made by local pathologists and therefore the differentiation between Burkitt lymphoma and Burkitt leukemia is not available for samples of the adult cohort as they are normally treated with the same protocol.

Cohort table – There is no stage reporting or LDH levels for the adult cases, again this should be included for consistency.

As stated above in detail, the samples comprising the adult cohort were obtained from different sources. Due to the nature of the sample collection clinical data were not available for the whole cohort. Therefore, corresponding analyses for this cohort are not possible.

Figure 1 – data should either be divided by Burkitt lymphoma and Burkitt leukaemia to make it clear what disease we are looking at. Also need to show that they are indeed similar at the genome level and can be combined for a prognostic study.

As outlined above we agree with the reviewer. As shown above, we have now carried out additional analyses comparing B-AL and Burkitt lymphoma (see new oncoplots for review above and Figure 4D of the revised manuscript). For clarification, we have added an additional line in Figure 1, indicating B-AL or Burkitt lymphoma cases.

Figure 2 – the genomes seem quite noisy for Burkitt lymphoma. They only include 72 pediatric cases, were they all Burkitt lymphoma?

We have clarified in the figure caption that this is an analysis for 55 BL and 17 B-AL cases combined.

Figure 5A is labelled as Overall survival (%) on Y-axis, but the figure legend only mentioned event-free survival.

We thank the reviewer for this important point and accordingly we have corrected the labeling (line 818).

Figure 6 – the plots are difficult to read and text is too small.

We agree with the reviewer, but due to space constraints the text is limited to the given text size. As most readers will read the article online, we will provide a high resolution figure where the text and the plots can be zoomed in for better visualization.

Supplementary Table 1 – should be separated by Lymphoma and leukaemia and clinical parameters such as LDH should be included.

The information on disease stages and LDH levels are shown in columns AF and AL of Supplementary Data 1.

Supplementary data 2 – Authors have included sample IDs in the somatic variant table but for ease of the reader an additional column stating which cohort the sample belongs to (BL/B-ALL) should be added, in addition to a column stating whether they are adult or pediatric cases.

As suggested by the reviewer, we have added patient age (column J) and BL/B-AL information (column K) in the revised version of Supplementary Data 2.

Supplementary data 4 – Authors comment on the number of sanger variants also detected by NGS, but fail to state the number of variants filtered out by their pipeline. Authors should also state in the text the number of variants called by their NGS pipeline that were not present/failed sanger.

As suggested, we have added this information in the text in the section “Validation by Sanger sequencing” on page 13 of the revised manuscript. To reduce the complexity of presentation of the validation results, the default view of Supplementary Data 4 mainly focuses on validation after filtering. However, this table also contains detailed information on prevented false positives (FPs) from DNA-sequencing as a result of the filtering hierarchy of our pipeline. To display them, please kindly remove the corresponding Excel filter on “validation class” (column AS). In the validation overview (third worksheet), these prevented FPs in DNA-sequencing due to filtering are aggregated: 272 in total. Precise filtering reasons for each variant are available (see second worksheet, column AH). The report of the three hard false positives from DNA-sequencing, i.e. remaining false positives relative to Sanger-sequencing after applying our filtering pipeline, can also be found in the text section “Validation by Sanger sequencing” (lines 452-481 of the revised manuscript).

Supplementary figure 6B – authors report importance of TP53 in patients treated with Rituximab but there are <5 events. Needs to be shown in a larger cohort before they can conclude this.

We agree with the reviewer’s comment that the relevance of TP53 mutations in patients who receive rituximab as part of their first-line treatment needs to be confirmed in larger series and trials. However, to date our reported series is the largest series available. For that reason, we decided to share the data even though the cohort is limited in size.

Some specific examples of grammatical errors.

Line 47 - “survival improved” are they missing a word? “Survival has improved”

Line 60 – “while it accounts for 1%”. What is ‘it’. The sentence should be phrased “in comparison BL accounts for ~1%” is a preferable phrase.

Line 67 – “recent reports have contributed to the molecular understanding of BL biology”. This is very vague and the authors need to elaborate on this.

Line 67 – The sentences beginning “In diffuse” is poorly worded and they don’t introduce why DLBCL is relevant in a BL paper.

Line 79 – “Molecular measurements” do they mean to say “Molecular markers”?

Line 90 – “Roughly 10%” is not very scientific. Suggest “Approximately 10%”.

In the term ‘p value’ the p should be in italics.

We thank the reviewer for detecting these mistakes. We have corrected them accordingly in the revised manuscript.

References

- 1 Grande, B. M. *et al.* Genome-wide discovery of somatic coding and noncoding mutations in pediatric endemic and sporadic Burkitt lymphoma. *Blood* **133**, 1313-1324, doi:10.1182/blood-2018-09-871418 (2019).
- 2 Newman, A. M. *et al.* Genomic abnormalities of TP53 define distinct risk groups of paediatric B-cell non-Hodgkin lymphoma. *Leukemia*, doi:10.1038/s41375-021-01444-6 (2021).
- 3 Panea, R. I. *et al.* The whole-genome landscape of Burkitt lymphoma subtypes. *Blood* **134**, 1598-1607, doi:10.1182/blood.2019001880 (2019).
- 4 Lopez, C. *et al.* Genomic and transcriptomic changes complement each other in the pathogenesis of sporadic Burkitt lymphoma. *Nature communications* **10**, 1459, doi:10.1038/s41467-019-08578-3 (2019).
- 5 Swerdlow, S. H. *et al.* The 2016 revision of the World Health Organization classification of lymphoid neoplasms. *Blood* **127**, 2375-2390, doi:10.1182/blood-2016-01-643569 (2016).
- 6 Swerdlow, S. H. *et al.* *WHO Classification of Tumors of Hematopoietic and Lymphoid Tissues*. 4th edn, (IARC Press, 2008).
- 7 Schmitz, R. *et al.* Burkitt lymphoma pathogenesis and therapeutic targets from structural and functional genomics. *Nature* **490**, 116-120, doi:10.1038/nature11378 (2012).
- 8 Zayac, A. S. *et al.* Outcomes of Burkitt lymphoma with central nervous system involvement: evidence from a large multicenter cohort study. *Haematologica* **106**, 1932-1942, doi:10.3324/haematol.2020.270876 (2021).
- 9 Fujita, N. *et al.* The role of hematopoietic stem cell transplantation with relapsed or primary refractory childhood B-cell non-Hodgkin lymphoma and mature B-cell leukemia: a retrospective analysis of enrolled cases in Japan. *Pediatr Blood Cancer* **51**, 188-192, doi:10.1002/pbc.21585 (2008).
- 10 Jourdain, A. *et al.* Outcome of and prognostic factors for relapse in children and adolescents with mature B-cell lymphoma and leukemia treated in three consecutive prospective "Lymphomes Malins B" protocols. A Societe Francaise des Cancers de l'Enfant study. *Haematologica* **100**, 810-817, doi:10.3324/haematol.2014.121434 (2015).
- 11 Cairo, M. S. *et al.* Advanced stage, increased lactate dehydrogenase, and primary site, but not adolescent age (≥ 15 years), are associated with an increased risk of treatment failure in children and adolescents with mature B-cell non-Hodgkin's lymphoma: results of the FAB LMB 96 study. *J Clin Oncol* **30**, 387-393, doi:10.1200/JCO.2010.33.3369 (2012).

REVIEWERS' COMMENTS

Reviewer #2 (Remarks to the Author):

The authors replied in an adequate way to all my comments, no further suggestions

Reviewer #3 (Remarks to the Author):

I appreciate the time taken to address our comments, and I believe the manuscript is much easier to read but still lacks novelty and confidence in the cohorts.

The authors still claim "This identifies the first molecular marker for relapse incidence in paediatric BL which will be used in future clinical trials." line which simply isn't true. There is one peer-reviewed paper (Newman et al) and one non-peer reviewed paper (Gong et al) that have already reported this (although I appreciate that the latter shouldn't be referenced until peer-reviewed). In addition, if this is the main finding there is still little discussion about it in the other sections of the paper, including the discussion.

I don't believe they have sufficiently addressed the collection of clinical data and pathology for the adult cohort to demonstrate that this result would be validated in a more uniformly collected and treated patient cohort. They argue that due to them coming from multiple centres, it is not possible to collect the information, however I think that this is key in particular in the adult cohort, where there is a higher risk of DLBCLs being included. I think that they should at least attempt to find this information and fill the gaps to ensure the cohort is representative.

One specific point - When they've applied the dN/dS analysis they don't seem to have applied multiple test correction? It seems odd that every gene analysed except TCF3 was identified to be significantly higher than 1? Are they claiming that 133 genes were identified as drivers in BL/B-AL? I think this is a major flaw of their analysis and they haven't addressed this point in the discussion, especially when Martincorena et al identified on average 1-4 driver substitutions per tumour. this work has the novelty required for nature comms.

Overall, the authors should be commended on their efforts addressing the comments, but I am concerned that the novelty has been lost due to recent important publications (Newman et al, Gong et al,). I think the key message is still muddy and difficult to pull out from the manuscript. I don't think this work has the novelty required for this journal.

Reviewer #4 (Remarks to the Author):

The revised manuscript has been improved according to the reviewer concerns. I suggest that it is suitable for publication.

REVIEWER COMMENTS

Reviewer #2 (Remarks to the Author):

The authors replied in an adequate way to all my comments, no further suggestions

We thank the reviewer for their reevaluation of our replies and their approval.

Reviewer #3 (Remarks to the Author):

I appreciate the time taken to address our comments, and I believe the manuscript is much easier to read but still lacks novelty and confidence in the cohorts.

The authors still claim "This identifies the first molecular marker for relapse incidence in paediatric BL which will be used in future clinical trials." line which simply isn't true. There is one peer-reviewed paper (Newman et al) and one non-peer reviewed paper (Gong et al) that have already reported this (although I appreciate that the latter shouldn't be referenced until peer-reviewed). In addition, if this is the main finding there is still little discussion about it in the other sections of the paper, including the discussion.

According to the suggestion of the reviewer, we revised the wording in line 57. Now it is stated there "This identifies a promising molecular marker for relapse incidence in pediatric BL which will be used in future clinical trials."

I don't believe they have sufficiently addressed the collection of clinical data and pathology for the adult cohort to demonstrate that this result would be validated in a more uniformly collected and treated patient cohort. They argue that due to them coming from multiple centres, it is not possible to collect the information, however I think that this is key in particular in the adult cohort, where there is a higher risk of DLBCLs being included. I think that they should at least attempt to find this information and fill the gaps to ensure the cohort is representative.

While we completely agree with the reviewer that it would be desirable to have the clinical information for patients of the adult cohort, we made every effort to collect these clinical data in a sufficient large fraction of investigated patients. Due to the retrospective character of our analysis, we unfortunately were not successful in collecting these clinical data. Thus, the desirable correlation with clinical parameters within the adult cohort is not possible within this project. We do not believe

that patients with the diagnosis of diffuse large B-cell lymphoma (DLBCL) are included in our analysis. Based on the histological and pathologic diagnosis and the immunophenotype, other aggressive lymphomas such as DLBCLs were excluded from our analysis.

One specific point - When they've applied the dN/dS analysis they don't seem to have applied multiple test correction? It seems odd that every gene analysed except TCF3 was identified to be significantly higher than 1? Are they claiming that 133 genes were identified as drivers in BL/B-AL? I think this is a major flaw of their analysis and they haven't addressed this point in the discussion, especially when Martincorena et al identified on average 1-4 driver substitutions per tumour. this work has the novelty required for nature comms.

As suggested by reviewer #1, we have applied dNdScv to test whether the genes depicted in our oncoplot in Figure 1 were positively selected by excess mutations above mutational background processes. We have performed multiple test correction using the Benjamini and Hochberg method over all tested hypotheses (oncoplot genes) for each respective subcohort (see column H in Supplementary Data 5). We only report confirmation of positive selection for the significant genes in this table (13 for the pediatric cohort); we do not make any such claim about all 133 genes in the panel. Importantly, this confirmatory analysis for oncoplot genes having $\geq 10\%$ cohort mutation frequency cannot be directly compared with an exploratory analysis based on pan-cancer whole exome sequencing. Specifically, regarding the count difference relative to the originally reported range of only 1-11 average drivers per entity (cf. Fig. 4B in Martincorena et al; notably, no lymphoma entity was included back then), we see reasons for potential differences in statistical power: genome-wide exploratory test versus our confirmatory test, a bias to potential cancer genes by design and purpose of our targeted panel, a higher coverage by targeted DNA-seq measurements relative to WES, and our fresh frozen sample quality that might have led to a lower noise rate for dNdScv's fit of background mutational processes.

Overall, the authors should be commended on their efforts addressing the comments, but I am concerned that the novelty has been lost due to recent important publications (Newman et al, Gong et al,). I think the key message is still muddy and difficult to pull out from the manuscript. I don't think this work has the novelty required for this journal.

One key message is the significant transition of the BL mutational profile over age that was neither revealed in the Newman et al nor in the Gong et al publication. Regarding TP53 results and notwithstanding a novelty defined by fastest publication, the scientific value of concurrently conducted independent studies leading to results that mutually confirm each other should not be undervalued, for example, independently discovered but overlapping clusters of genetic lesions in DLBCL (compare Wright et al in Cancer Cell 2020 and Chapuy et al in Nature Medicine 2018).

Reviewer #4 (Remarks to the Author):

The revised manuscript has been improved according to the reviewer concerns. I suggest that it is suitable for publication

We thank the reviewer for their evaluation and approval.